psychology

affect rating, disgust, fear, open-access

**Author for correspondence:**
Risako Shirai
e-mail: risakoshirai@gmail.com

# Open biological negative image set

## Risako Shirai[1,2] and Katsumi Watanabe[1,3]

[1]Faculty of Science and Engineering, Waseda University, Postal code: 3-4-1 Okubo, Shinjuku-ku, Tokyo, 169-8555, Japan
[2]Japan Society for the Promotion of Science, Tokyo, Japan
[3]University of New South Wales, Kensington, Australia

RS, 0000-0003-1489-7106

Scientists conducting affective research often use visual, emotional images, to examine the mechanisms of defensive responses to threatening and dangerous events and objects. Many studies use the rich emotional images from the International Affective Picture System (IAPS) to facilitate affective research. While IAPS images can be classified into emotional categories such as fear or disgust, the number of images per discrete emotional category is limited. We developed the Open Biological Negative Image Set (OBNIS) consisting of 200 colour and greyscale creature images categorized as disgusting, fearful or neither. Participants in Experiment 1 ($N = 210$) evaluated the images' valence and arousal and classified them as *disgusting*, *fearful* or *neither*. In Experiment 2, other participants ($N = 423$) rated the disgust and fear levels of the images. As a result, the OBNIS provides valence, arousal, disgust and fear ratings and 'disgusting,' 'fearful' and 'neither' emotional categories for each image. These images are available to download on the Internet (https://osf.io/pfrx4/?view_only=911b1be722074ad4aab87791cb8a72f5).

# 1. Introduction

Human beings experience various emotions. In particular, negative emotions such as fear and disgust are essential mechanisms for avoiding dangerous objects and events in our daily lives. Moreover, strong negative emotions also cause phobias. Research has suggested that fear and anxiety play essential roles in causing and maintaining phobias (for review, [1]). More recently, it has been suggested that disgust and fear play crucial roles in causing specific phobias. For example, Olatunji *et al.* [2] reported that disgust and fear interact in predicting blood injection phobia [2]. Also, many studies have reported that disgust of spiders is the primary emotion in spider phobia ([3–5], but not [6–9]).

Several behavioural and neuroscience studies designed to evaluate defensive responses to threatening and dangerous stimuli have used various types of negative images, including images of snakes, traffic accidents, spiders and dirty toilets. These negative images are catalogued in rich image databases that

contain a large amount of image data. Such databases include the International Affective Picture System (IAPS), a standardized set of emotional images [10]; the Geneva Affective Picture Database, another standardized collection of emotional images [11]; the Nencki Affective Picture System (NAPS), an image set of realistic and high-quality emotional images [12]; the basic-emotion normative ratings for the Nencki Affective Picture System, a standardized image set of NAPS images with additional information on discrete emotions [13]; the Open Affective Standardized Image Set, an open-access set of online emotional images [14]; the Set of Fear Inducing Pictures, an image set of phobia-relevant pictures [15]; the DIsgust-RelaTed-Images database (DIRTI), a standardized set of disgusting pictures [16].

The IAPS has been widely used in affective research, and many studies have relied on this database. However, since the IAPS is based on the dimensional approach to emotions and focuses on valence and arousal [12,14], it does not map onto discrete emotions such as disgust [16]. Although several studies have classified the IAPS images into discrete emotional categories (e.g. [17–19]), the number of images per emotion is somewhat limited, especially for disgust. Haberkamp et al. [16] has provided 240 disgust-related images in the DIRTI (e.g. animal carcasses, rotten food and blood) for researchers interested in studying different facets of disgust. However, the DIRTI database provides only disgust-related images. Thus, researchers interested in comparing the effects of discrete emotions such as fear and disgust must gather images from several image databases, making it difficult to select images based on a standardized rating.

Also, the images of most databases do not contain complete objects (e.g. an image shows a snake's face but not the whole body). Additionally, researchers often remove unnecessary backgrounds from images and use the trimmed object as experimental stimuli to ensure a pure effect (e.g. [20,21]). Almost none of the image databases, including the IAPS, have isolated, full object images. Therefore, many researchers in affective science need to collect object images from scratch. In particular, although some image databases provide images of disgusting objects (e.g. images in the DIRTI, [16]), many disgusting images cause disgust through an understanding of the context of the whole scene (e.g. dirty toilets, dead animals) and thus, the total number of disgusting object images is quite small.

The current study's goal was to develop a database of negative emotional objects to address the issues with current databases and reduce the need for detailed, time-consuming and labour-intensive image adjustments. We collected 200 images depicting negative and neutral objects from open-access resources by extracting the images of the object from the background scene. In particular, the current study focused on images associated with disgust and fear and used images that could evoke these emotions by themselves in developing an image database. This database was expected to facilitate investigations of the effects of disgust and fear on behaviour, including phobias. The participants were asked to rate these images in terms of affective values. The images were presented in greyscale or colour. The reason for utilizing both colour and greyscale images was that none of the current databases provide rating scores for the greyscale images. Greyscale images are often required in experiments in which the physical characteristics of images must be controlled. The participants reported the valence and arousal scores for the colour or greyscale images and categorized each image as disgusting, fearful or neither. We calculated mean values and standard deviations of valence and arousal ratings for each image. The probability of classifying each image into *disgusting*, *fearful* or *neither* categories was calculated using each participant's image categorization.

Moreover, we focused on the extent to which people could distinguish between disgust and fear in each image. For example, several studies have suggested that people have culturally acquired disgust of specific animals and insects associated with contamination and disease, such as spiders, and these animals and insects have become fear-relevant [22,23]. Moreover, Woody & Teachman [24] reported that people tend to confuse disgust and fear when the intensity of elicited emotions is moderate [24,25]. Therefore, we expected that asking participants to label many visual images as 'disgusting' or 'fearful' would allow assessing the extent to which fear and disgust are combined in a particular image and how precisely people could distinguish negativity into the two categorical emotions of fear and disgust. We performed cluster analysis on each image's probability of being classified as 'disgusting' or 'fearful' and classified each image by the contents of their emotions.

# 2. Experiment 1

## 2.1. Methods

### 2.1.1. Participants

Participants were recruited through the crowdsourcing services (Crowd Works) to rate images for payment of 300 yen. Two-hundred and ten workers (112 females, 98 males, mean age = 40.31 years, age range = 20–68

years) participated. We used a crowdsourcing service in Japan. Therefore, we consider that nearly all the participants in this study were Japanese people or could understand Japanese. All participants gave their informed consent online before participating in the study. Ethical approval for the study was obtained from the Waseda University's Ethics Review Committee on Research with Human Subjects.

### 2.1.2. Materials

Two-hundred images were collected from Google Images (https://images.google.com). All images were chosen to represent emotionally negative (e.g. snakes, wasps, spiders and cockroaches) or neutral objects (e.g. cats, flowers). The image search was limited to images classified as available for reuse with modification. We trimmed objects from images of scenes and resized them to $500 \times 500$ pixels in Adobe Photoshop CC 2015. All images were altered to the greyscale. Finally, 400 images consisting of 200 colour and greyscale images with 100 emotionally negative and 100 emotionally neutral images were prepared for the study.

### 2.1.3. Procedure

The participants were randomly assigned to one of two groups. All images were presented in colour (colour condition; $N = 100$) to one group, and all images were presented in greyscale (greyscale condition; $N = 110$) to the other. The 200 images were randomly presented in sequence on a browser window using the Qualtrics platform (Qualtrics, Provo, UT). Participants were instructed to rate each image on valence and arousal dimensions (See appendix A). Valence was assessed by a 9-point scale ranging from 1 (*extremely negative*) to 9 (*extremely positive*). Arousal was assessed by a 9-point scale ranging from 1 (*low arousal*) to 9 (*high arousal*). Also, participants were instructed to categorize each image as '*disgusting*', '*fearful*' or '*neither*', according to whether they predominantly felt disgusted or fearful when seeing the image. Participants were asked to select '*neither*' if they felt neither disgust nor fear. Each image was kept presented on the window until all the responses were obtained. It took approximately 45 min to complete the experimental session.

## 2.2. Results and discussion

### 2.2.1. Univariate distributions

Figure 1 shows the relationship between means and standard deviations of valence and arousal ratings. Visual inspection indicates that mean valence variability and arousal ratings for colour and greyscale conditions were normally distributed. Each image's valence rating ranged from 1.70 to 7.63 in the colour condition and from 1.72 to 7.56 in the greyscale condition. The mean valence rating was 4.40 in the colour condition and 4.32 in the greyscale condition. The results indicate that the valence rating range's midpoint was lower than the theoretical midpoint of the scale for each presentation mode. The arousal rating per image ranged from 2.66 to 6.94 in the colour condition and from 2.22 to 6.92 in the greyscale condition. The mean arousal rating was 4.84 in the colour condition and 4.67 in the greyscale condition.

We evaluated the face validity of the valence and arousal ratings for the images by probing the images with the highest positive valence, highest negative valence, the highest arousal ratings and lowest arousal ratings in each mode of the presentation. In the colour condition, the image with the most positive valence was that of a cat (image no. 108; mean = 7.63, s.d. = 1.46), and the image with the most negative valence was that of a cockroach (image no. 1; mean = 1.70, s.d. = 1.14). The image with the highest arousal was that of a snake (image no. 60; mean = 6.94, s.d. = 2.30), and the image with the lowest arousal was that of a leaf (image no. 119; mean = 2.66, s.d. = 1.76). In the greyscale condition, the image with the highest positive valence was that of a cat (image no. 108; mean = 7.55, s.d. = 1.44), and the image with the most negative valence (mean = 1.72, s.d. = 0.94) and the most arousing image was that of a centipede (image no. 23; mean = 6.92, s.d. = 2.13). The least arousing image was that of a leaf (image no. 121; mean = 2.22, s.d. = 1.62). These images established the face validity of the valence and arousal scores because they fit the ratings.

### 2.2.2. Relationships between valence and arousal

Next, we examined the relationship between valence and arousal ratings for images in each presentation mode. We found that the mean valence ratings of images were negatively correlated with the mean

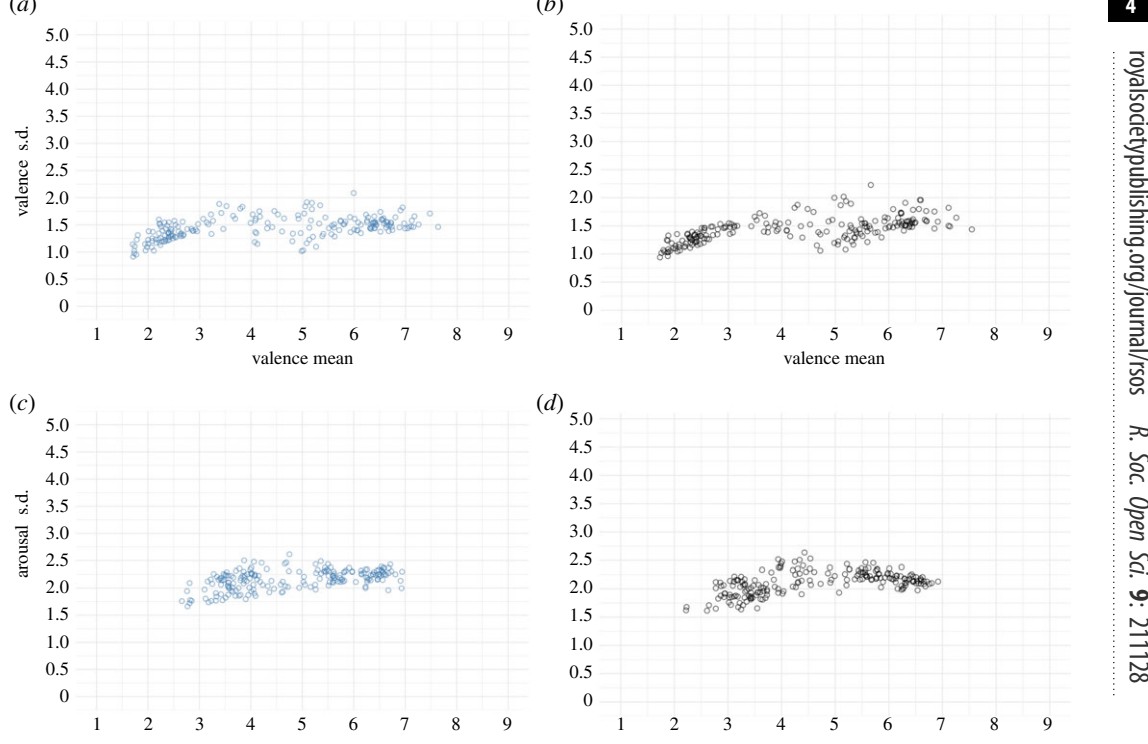

**Figure 1.** Relationship between the mean value and standard deviation of the valence (*a,b*) and arousal ratings (*c,d*). (*a,c*) show the colour condition. (*b,d*) display the greyscale condition.

arousal ratings of images in both presentation modes (colour condition: $r = -0.87$ ($-0.90$; $-0.83$), $t_{198} = -24.41$, $p < 0.001$; greyscale condition: $r = -0.85$ ($-0.88$; $-0.81$), $t_{198} = -22.71$, $p < 0.001$). The Open Biological Negative Image Set (OBNIS) contains both emotionally negative and neutral (or relatively positive) images. Since there appears to be no correlation for the neutral valence images in the OBNIS from visual inspections, the strong negative correlation between valence and arousal ratings for images seen in this study can be attributed more to the emotionally negative images in the OBNIS.

### 2.2.2.1. Gender differences

Previous studies suggest gender differences in affective processing [26–28]. Therefore, we explored whether gender differences modulated valence and arousal ratings. Figure 2 shows the relationship between valence and arousal ratings by gender. A correlation analysis indicated that images' valence ratings were negatively correlated with arousal ratings in both genders (women: $r = -0.90$ ($-0.92$; $-0.87$), $t_{198} = -29.17$, $p < 0.001$; men: $r = -0.79$ ($-0.83$; $-0.73$), $t_{198} = -17.88$, $p < 0.001$). However, the correlation between valence and arousal ratings was stronger in women than men for the colour condition ($z = 4.12$, $p < 0.001$). These results, similar to Bradley *et al.* [26], indicate that gender modulated the relationship between images' valence and arousal ratings. Conversely, the correlations between valence and arousal ratings in the greyscale condition did not differ between genders (women: $r = -0.85$ ($-0.88$; $-0.80$), $t_{198} = -22.56$, $p < 0.001$; men: $r = -0.84$ ($-0.88$; $-0.80$), $t_{198} = -21.98$, $p < 0.001$; $z = 0.22$, $p = 0.83$). These findings indicate that the effect of gender on the relationship between valence and arousal depends on whether an image is presented in colour or not. The majority of the participants in this study were Japanese. Therefore, the extent to which the finding of this study applies to samples from other countries and cultures should be examined in the future.

### 2.2.3. Emotion categorization

We calculated each image's probability of being classified into each category (i.e. *disgust*, *fear* or *neither*) using cluster analysis with NbClust [29]. NbClust is a software package in R that provides 30 indices for identifying the optimal number of clusters, which proposes using the optimum number of clusters based on the majority rule. In the present study, the NbClust analysis was conducted on 200 images in each presentation mode for the three variables (the probabilities of categorizing as *disgust*, *fear* or *neither*)

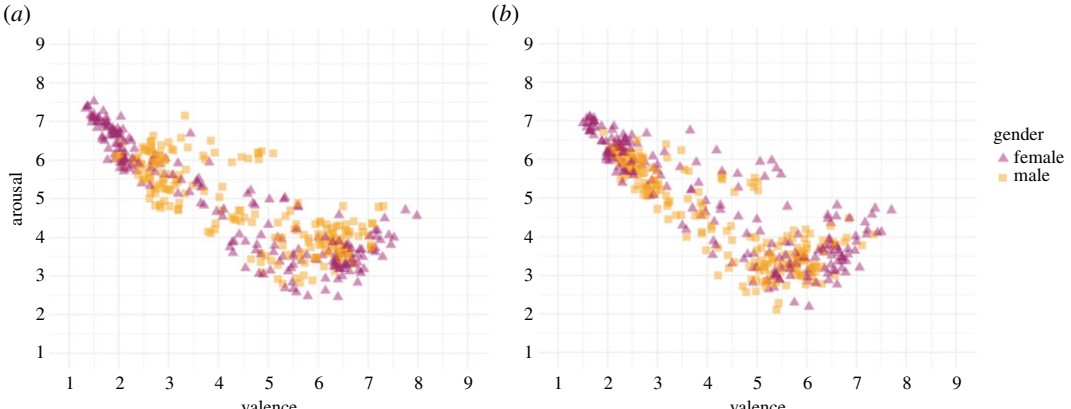

**Figure 2.** Relationship between valence and arousal ratings by gender. (*a*) shows the colour condition. (*b*) displays the greyscale condition.

using the 'ward.D2' method [30]. The results show that the three clusters were derived from each cluster analysis in the two presentation conditions. Table 1 displays the probability of assigning each image category within each cluster. Cluster 1 in the colour condition was named the 'disgust-related group' because the most frequently labelled category for images within this cluster was '*disgust*' (figure 3; *disgust mean ratio: 61.70%, fear mean ratio: 18.52%, neither mean ratio: 19.79%*). Cluster 2 was named the 'fear-related group', since the most frequently labelled category for images within this cluster was '*fear*' (figure 3; *disgust mean ratio: 15.68%, fear mean ratio: 64.05%, neither mean ratio: 20.27%*). Cluster 3 was named the 'neither group' because the most frequently labelled category for images within this cluster was '*neither*' (figure 3; *disgust mean ratio: 4.05%, fear mean ratio: 1.62%, neither mean ratio: 94.32%*). Cluster 1 in the greyscale condition was named the 'disgust-related group' (figure 5; *disgust mean ratio: 62.55%, fear mean ratio: 15.51%, neither mean ratio: 21.95%*), cluster 2 was named the 'fear-related group' (figure 3; *disgust mean ratio: 15.87%, fear mean ratio: 62.21%, neither mean ratio: 21.91%*) and cluster 3 was named the 'neither group' (figure 3; *disgust mean ratio: 5.10%, fear mean ratio: 1.84%, neither mean ratio: 93.07%*) using the same strategy as in the colour condition. The images assigned to the disgust-related and fear-related groups were judged as *disgusting* or *fearful*, with a relatively high probability across participants. Thus, the participants were able to distinguish the emotion they experienced between 'disgust' and 'fear' in response to the images relatively well. However, some images in the disgust-related group were assigned with almost the same probability to 'disgust' and 'fear' (i.e. the difference is only 10% or less; e.g. the images in colour condition: Image ID 9, 27, 7, 30, 55, 28, 48, 49, 97, 75 and 74; the images in greyscale condition: Image ID 6, 30, 28, 49, 74, 75 and 97). Thus, it should be noted that the participants might confuse the feelings of disgust and fear when viewing such images.

We also examined whether the images assigned to any of the three categories differed between colour and greyscale conditions (table 2). The results indicated that 70 images were assigned to the disgust-related group in the greyscale condition. By contrast, 65 of the identical images were assigned to the disgust-related group in the colour condition, two of the identical images were assigned to the fear-related group and three of the identical images were assigned to the neither group. The 39 images assigned to the fear-related group in the greyscale condition were all assigned to the colour condition's fear-related group. Ninety-one images were assigned to the neither group in the greyscale condition, whereas 90 of the identical images were assigned to the neither group in the colour condition, and one identical image was assigned to the disgust-related group. These results indicated that each subgroup of greyscale images was very similar to images in the colour condition's respective subgroup.

### 2.2.3.1. Valence and arousal ratings

Studies have suggested that emotional valence and arousal levels are different among discrete emotional categories of images [31–34]. Thus, we calculated the mean valence and arousal ratings per subgroup (disgust-related, fear-related and neither groups) in colour and greyscale to identify whether the valence and arousal ratings were different among the subgroups of images. First, the mean valence ratings for coloured images were subjected to a one-way analysis of variance (ANOVA) with the image groups (disgust-related, fear-related and neither groups) as independent variables and the valence

**Table 1.** Each table lists the valence, the arousal, and the probabilities of assigning images in each cluster to 'disgust', 'fear' or 'neither' categories. Image ID refers to an ID assigned to each image. The closer was the colour in the cell to white; the closer the probability of the image being assigned to categories 'disgust', 'fear' or 'neither' was to 0%. The closer was the colour in the cell to red, the closer the probability of the image being assigned to categories.

| Cluster 1 in colour condition: Disgust-related group | | | | | | Cluster 1 in greyscale condition: Disgust-related group | | | | | |
|---|---|---|---|---|---|---|---|---|---|---|---|
| Image ID | Valence | Arousal | Disgust ratio | Fear ratio | Neither ratio | Image ID | Valence | Arousal | Disgust ratio | Fear ratio | Neither ratio |
| 21 | 2.34 | 5.46 | 87.00 | 7.00 | 6.00 | 4 | 2.09 | 5.77 | 93.64 | 1.82 | 4.55 |
| 31 | 2.11 | 5.97 | 85.00 | 9.00 | 6.00 | 32 | 2.07 | 6.24 | 90.00 | 5.45 | 4.55 |
| 35 | 2.31 | 5.97 | 85.00 | 8.00 | 7.00 | 13 | 1.85 | 6.60 | 89.09 | 7.27 | 3.64 |
| 22 | 1.71 | 6.57 | 84.00 | 11.00 | 5.00 | 22 | 1.87 | 6.51 | 89.09 | 7.27 | 3.64 |
| 39 | 2.38 | 5.50 | 84.00 | 9.00 | 7.00 | 47 | 2.14 | 6.24 | 89.09 | 8.18 | 2.73 |
| 41 | 2.27 | 5.61 | 84.00 | 8.00 | 8.00 | 12 | 1.86 | 6.65 | 87.27 | 9.09 | 3.64 |
| 43 | 2.44 | 5.55 | 84.00 | 6.00 | 10.00 | 31 | 2.19 | 5.89 | 85.45 | 7.27 | 7.27 |
| 44 | 2.48 | 5.26 | 84.00 | 7.00 | 9.00 | 45 | 2.07 | 6.24 | 85.45 | 11.82 | 2.73 |
| 4 | 2.40 | 5.37 | 82.00 | 8.00 | 10.00 | 33 | 2.54 | 5.59 | 84.55 | 3.64 | 11.82 |
| 12 | 1.74 | 6.71 | 82.00 | 15.00 | 3.00 | 40 | 2.47 | 5.26 | 84.55 | 2.73 | 12.73 |
| 23 | 2.54 | 5.51 | 82.00 | 10.00 | 8.00 | 1 | 1.78 | 6.68 | 83.64 | 12.73 | 3.64 |
| 38 | 2.34 | 5.85 | 82.00 | 11.00 | 7.00 | 39 | 2.37 | 5.65 | 83.64 | 6.36 | 10.00 |
| 1 | 1.70 | 6.73 | 81.00 | 16.00 | 3.00 | 41 | 2.42 | 5.52 | 83.64 | 5.45 | 10.91 |
| 20 | 2.22 | 5.91 | 81.00 | 9.00 | 10.00 | 5 | 2.91 | 4.91 | 82.73 | 4.55 | 12.73 |
| 32 | 1.94 | 6.48 | 81.00 | 11.00 | 8.00 | 43 | 2.47 | 5.58 | 82.73 | 3.64 | 13.64 |
| 19 | 2.39 | 5.74 | 80.00 | 8.00 | 12.00 | 44 | 2.58 | 5.44 | 82.73 | 6.36 | 10.91 |
| 24 | 1.73 | 6.54 | 80.00 | 13.00 | 7.00 | 20 | 2.28 | 5.81 | 81.82 | 8.18 | 10.00 |
| 17 | 2.07 | 6.15 | 79.00 | 16.00 | 5.00 | 34 | 1.87 | 6.43 | 81.82 | 12.73 | 5.45 |
| 34 | 1.78 | 6.59 | 79.00 | 17.00 | 4.00 | 15 | 2.01 | 6.29 | 80.91 | 17.27 | 1.82 |
| 40 | 2.63 | 5.27 | 79.00 | 6.00 | 15.00 | 18 | 1.75 | 6.60 | 80.91 | 18.18 | 0.91 |
| 33 | 2.48 | 5.42 | 78.00 | 4.00 | 18.00 | 19 | 2.17 | 5.93 | 80.91 | 11.82 | 7.27 |
| 13 | 1.73 | 6.58 | 77.00 | 21.00 | 2.00 | 24 | 2.01 | 6.38 | 80.91 | 12.73 | 6.36 |
| 47 | 2.08 | 6.33 | 77.00 | 14.00 | 9.00 | 35 | 2.20 | 5.98 | 80.91 | 10.00 | 9.09 |
| 11 | 2.24 | 6.04 | 76.00 | 18.00 | 6.00 | 17 | 1.86 | 6.45 | 80.00 | 19.09 | 0.91 |
| 5 | 2.84 | 5.05 | 75.00 | 3.00 | 22.00 | 37 | 2.40 | 5.48 | 79.09 | 9.09 | 11.82 |
| 37 | 2.41 | 5.36 | 75.00 | 13.00 | 12.00 | 42 | 2.79 | 5.68 | 79.09 | 9.09 | 11.82 |
| 42 | 2.69 | 5.73 | 75.00 | 13.00 | 12.00 | 11 | 2.23 | 6.16 | 76.36 | 12.73 | 10.91 |
| 45 | 2.15 | 6.23 | 73.00 | 22.00 | 5.00 | 14 | 1.80 | 6.53 | 75.45 | 23.64 | 0.91 |
| 10 | 2.64 | 5.60 | 70.00 | 15.00 | 15.00 | 38 | 2.32 | 5.82 | 75.45 | 12.73 | 11.82 |
| 15 | 1.95 | 6.33 | 70.00 | 27.00 | 3.00 | 21 | 2.30 | 5.78 | 74.55 | 12.73 | 12.73 |
| 18 | 2.00 | 6.47 | 70.00 | 22.00 | 8.00 | 10 | 2.43 | 5.85 | 71.82 | 17.27 | 10.91 |
| 54 | 2.64 | 5.53 | 69.00 | 22.00 | 9.00 | 54 | 2.55 | 5.51 | 71.82 | 14.55 | 13.64 |
| 16 | 1.80 | 6.81 | 67.00 | 33.00 | 0.00 | 16 | 1.72 | 6.92 | 70.91 | 28.18 | 0.91 |
| 14 | 1.77 | 6.61 | 66.00 | 31.00 | 3.00 | 23 | 3.08 | 4.99 | 69.09 | 9.09 | 21.82 |
| 26 | 2.16 | 6.14 | 66.00 | 27.00 | 7.00 | 26 | 2.05 | 6.45 | 66.36 | 30.91 | 2.73 |
| 92 | 2.68 | 5.30 | 66.00 | 17.00 | 17.00 | 8 | 1.91 | 6.65 | 63.64 | 31.82 | 4.55 |
| 93 | 2.76 | 5.44 | 61.00 | 17.00 | 22.00 | 94 | 2.71 | 5.46 | 62.73 | 15.45 | 21.82 |
| 25 | 2.00 | 6.57 | 60.00 | 31.00 | 9.00 | 55 | 3.03 | 5.21 | 60.91 | 16.36 | 22.73 |
| 94 | 2.45 | 5.82 | 60.00 | 29.00 | 11.00 | 2 | 3.49 | 5.07 | 59.09 | 8.18 | 32.73 |
| 8 | 2.09 | 6.38 | 58.00 | 37.00 | 5.00 | 25 | 2.04 | 6.63 | 59.09 | 36.36 | 4.55 |
| 95 | 3.23 | 4.72 | 53.00 | 13.00 | 34.00 | 27 | 1.87 | 6.80 | 59.09 | 37.27 | 3.64 |
| 50 | 3.65 | 5.25 | 52.00 | 11.00 | 37.00 | 93 | 2.80 | 5.26 | 59.09 | 14.55 | 26.36 |
| 9 | 2.20 | 6.46 | 50.00 | 41.00 | 9.00 | 9 | 1.96 | 6.59 | 57.27 | 39.09 | 3.64 |
| 29 | 2.92 | 5.60 | 50.00 | 26.00 | 24.00 | 92 | 3.11 | 4.86 | 56.36 | 7.27 | 36.36 |
| 27 | 1.98 | 6.50 | 49.00 | 47.00 | 4.00 | 193 | 3.86 | 4.55 | 53.64 | 0.91 | 45.45 |
| 36 | 3.99 | 4.67 | 49.00 | 7.00 | 44.00 | 87 | 2.60 | 5.55 | 52.73 | 30.91 | 16.36 |
| 2 | 3.52 | 5.34 | 48.00 | 14.00 | 38.00 | 7 | 1.85 | 6.75 | 51.82 | 41.82 | 6.36 |
| 106 | 3.26 | 5.26 | 46.00 | 28.00 | 26.00 | 95 | 3.66 | 4.21 | 50.91 | 6.36 | 42.73 |
| 81 | 2.88 | 5.51 | 45.00 | 32.00 | 23.00 | 50 | 3.71 | 4.93 | 50.00 | 9.09 | 40.91 |
| 193 | 4.17 | 5.02 | 45.00 | 9.00 | 46.00 | 6 | 1.94 | 6.59 | 49.09 | 46.36 | 4.55 |
| 7 | 2.01 | 6.91 | 44.00 | 51.00 | 5.00 | 81 | 3.01 | 5.56 | 46.36 | 32.73 | 20.91 |
| 30 | 2.86 | 5.79 | 43.00 | 34.00 | 23.00 | 106 | 3.43 | 4.30 | 46.36 | 13.64 | 40.00 |
| 55 | 3.17 | 5.61 | 43.00 | 39.00 | 18.00 | 48 | 2.85 | 5.67 | 45.45 | 35.45 | 19.09 |
| 28 | 2.95 | 5.59 | 41.00 | 33.00 | 26.00 | 36 | 3.88 | 4.37 | 44.55 | 8.18 | 47.27 |
| 48 | 2.97 | 5.78 | 41.00 | 44.00 | 15.00 | 29 | 3.11 | 5.36 | 42.73 | 27.27 | 30.00 |
| 191 | 4.03 | 5.03 | 40.00 | 5.00 | 55.00 | 30 | 2.94 | 5.76 | 40.91 | 39.09 | 20.00 |
| 3 | 4.16 | 4.94 | 37.00 | 7.00 | 56.00 | 191 | 3.96 | 4.61 | 40.91 | 3.64 | 55.45 |
| 49 | 3.14 | 5.69 | 36.00 | 42.00 | 22.00 | 189 | 4.12 | 4.31 | 39.09 | 2.73 | 58.18 |
| 105 | 4.08 | 4.07 | 36.00 | 10.00 | 54.00 | 105 | 3.90 | 3.94 | 37.27 | 15.45 | 47.27 |
| 189 | 4.09 | 4.62 | 34.00 | 5.00 | 61.00 | 28 | 3.03 | 5.59 | 36.36 | 37.27 | 26.36 |
| 46 | 4.32 | 4.62 | 32.00 | 7.00 | 61.00 | 49 | 3.05 | 5.41 | 36.36 | 40.00 | 23.64 |
| 109 | 4.10 | 3.92 | 32.00 | 12.00 | 56.00 | 46 | 4.53 | 4.37 | 34.55 | 2.73 | 62.73 |
| 107 | 4.07 | 4.01 | 27.00 | 16.00 | 57.00 | 3 | 4.35 | 4.53 | 32.73 | 9.09 | 58.18 |
| 97 | 4.12 | 4.60 | 24.00 | 22.00 | 54.00 | 108 | 4.15 | 3.61 | 32.73 | 7.27 | 60.00 |
| 75 | 4.31 | 4.18 | 23.00 | 23.00 | 54.00 | 114 | 4.38 | 3.18 | 32.73 | 2.73 | 64.55 |
| 74 | 4.41 | 4.40 | 18.00 | 23.00 | 59.00 | 109 | 4.09 | 3.73 | 27.27 | 11.82 | 60.91 |
| | | | | | | 192 | 4.87 | 4.04 | 27.27 | 2.73 | 70 |
| | | | | | | 74 | 4.68 | 3.70 | 18.18 | 13.64 | 68.18 |
| | | | | | | 75 | 4.72 | 3.55 | 18.18 | 12.73 | 69.09 |
| | | | | | | 97 | 4.09 | 4.73 | 17.27 | 31.82 | 50.91 |

Ratio

0 % ▭▭▭▭▭ 100%

(Continued.)

ratings as the dependent variable. The main effect of image group was significant, $F_{2,197} = 408.07$, $p < 0.001$, $\eta_p^2 = 0.81$. *Post hoc* comparisons using the Holm methods indicated that the images in the neither group (mean = 6.15, s.d. = 0.69) were rated as less negative than the images in the disgust-related group (mean = 2.72, s.d. = 0.81; $t_{197} = 26.25$, $p < 0.001$, $d = 4.53$) or the fear-related group (mean = 3.13, s.d. = 1.03; $t_{197} = 19.89$, $p < 0.001$, $d = 3.45$). Moreover, the images in the disgust-related group were rated as more negative than the images in the fear-related group, $t_{197} = 2.50$, $p = 0.01$, $d = 0.44$. These results indicated that the valence ratings were different among colour image subgroups. We also observed the main effect of image group on the mean valence ratings for greyscale images, $F_{2,197} = 346.98$, $p < 0.001$, $\eta_p^2 = 0.78$. Similar to the findings of colour images, the greyscaled images in the neither group

| Cluster 2 in colour condition: Fear-related group | | | | | | Cluster 2 in greyscale condition: Fear-related group | | | | |
|---|---|---|---|---|---|---|---|---|---|---|
| Image ID | Valence | Arousal | Disgust ratio | Fear ratio | Neither ratio | Image ID | Valence | Arousal | Disgust ratio | Fear ratio | Neither ratio |
| 53 | 3.38 | 6.93 | 0.00 | 90.00 | 10.00 | 51 | 2.49 | 6.33 | 12.73 | 80.91 | 6.36 |
| 96 | 3.22 | 6.26 | 2.00 | 86.00 | 12.00 | 53 | 3.58 | 6.54 | 1.82 | 80.00 | 18.18 |
| 51 | 2.51 | 6.63 | 12.00 | 80.00 | 8.00 | 96 | 2.95 | 6.55 | 3.64 | 80.00 | 16.36 |
| 68 | 2.40 | 6.56 | 5.00 | 76.00 | 19.00 | 76 | 2.32 | 6.71 | 13.64 | 78.18 | 8.18 |
| 76 | 2.23 | 6.68 | 14.00 | 76.00 | 10.00 | 62 | 2.64 | 6.26 | 20.91 | 72.73 | 6.36 |
| 84 | 3.07 | 6.23 | 21.00 | 76.00 | 3.00 | 68 | 3.16 | 6.13 | 11.82 | 70.91 | 17.27 |
| 65 | 3.83 | 6.23 | 3.00 | 75.00 | 22.00 | 65 | 4.24 | 5.97 | 3.64 | 69.09 | 27.27 |
| 83 | 2.34 | 6.37 | 22.00 | 74.00 | 4.00 | 84 | 2.28 | 6.51 | 27.27 | 69.09 | 3.64 |
| 73 | 2.59 | 6.67 | 22.00 | 73.00 | 5.00 | 73 | 2.37 | 6.34 | 27.27 | 68.18 | 4.55 |
| 89 | 3.79 | 5.76 | 3.00 | 73.00 | 24.00 | 69 | 3.53 | 5.75 | 2.73 | 67.27 | 30.00 |
| 91 | 2.44 | 6.45 | 1.00 | 72.00 | 27.00 | 83 | 2.40 | 6.30 | 22.73 | 67.27 | 10.00 |
| 99 | 3.69 | 5.73 | 19.00 | 72.00 | 9.00 | 52 | 2.55 | 6.45 | 25.45 | 66.36 | 8.18 |
| 69 | 3.42 | 5.82 | 7.00 | 71.00 | 22.00 | 58 | 2.55 | 6.27 | 20.00 | 65.45 | 14.55 |
| 52 | 2.21 | 6.94 | 25.00 | 69.00 | 6.00 | 71 | 4.29 | 5.62 | 0.91 | 64.55 | 34.55 |
| 58 | 2.67 | 6.39 | 21.00 | 67.00 | 12.00 | 78 | 2.24 | 6.40 | 26.36 | 64.55 | 9.09 |
| 100 | 2.40 | 6.30 | 22.00 | 67.00 | 11.00 | 88 | 2.29 | 6.22 | 29.09 | 63.64 | 7.27 |
| 78 | 2.28 | 6.54 | 23.00 | 65.00 | 12.00 | 89 | 3.68 | 5.85 | 3.64 | 63.64 | 32.73 |
| 88 | 2.24 | 6.43 | 29.00 | 64.00 | 7.00 | 56 | 2.42 | 6.15 | 22.73 | 62.73 | 14.55 |
| 56 | 2.32 | 6.56 | 24.00 | 63.00 | 13.00 | 99 | 2.55 | 6.01 | 24.55 | 62.73 | 12.73 |
| 60 | 2.22 | 6.54 | 26.00 | 63.00 | 11.00 | 100 | 2.67 | 6.04 | 18.18 | 62.73 | 19.09 |
| 77 | 2.21 | 6.49 | 22.00 | 63.00 | 15.00 | 79 | 2.28 | 6.46 | 28.18 | 61.82 | 10.00 |
| 62 | 2.33 | 6.27 | 25.00 | 62.00 | 13.00 | 91 | 3.75 | 5.18 | 5.45 | 60.91 | 33.64 |
| 82 | 2.40 | 6.27 | 28.00 | 62.00 | 10.00 | 77 | 2.39 | 6.08 | 27.27 | 60.00 | 12.73 |
| 85 | 2.44 | 6.15 | 32.00 | 62.00 | 6.00 | 57 | 2.26 | 6.30 | 30.00 | 59.09 | 10.91 |
| 86 | 2.58 | 6.26 | 32.00 | 62.00 | 6.00 | 60 | 2.50 | 6.31 | 25.45 | 59.09 | 15.45 |
| 6 | 2.08 | 6.68 | 33.00 | 61.00 | 6.00 | 72 | 4.48 | 5.56 | 0.91 | 59.09 | 40.00 |
| 57 | 2.25 | 6.52 | 29.00 | 61.00 | 10.00 | 90 | 3.93 | 4.90 | 2.73 | 59.09 | 38.18 |
| 71 | 4.52 | 5.62 | 1.00 | 61.00 | 38.00 | 70 | 4.59 | 5.22 | 0.91 | 58.18 | 40.91 |
| 80 | 2.46 | 6.20 | 23.00 | 60.00 | 17.00 | 80 | 2.45 | 6.05 | 26.36 | 57.27 | 16.36 |
| 72 | 4.45 | 5.59 | 1.00 | 59.00 | 40.00 | 85 | 2.36 | 6.15 | 31.82 | 56.36 | 11.82 |
| 79 | 2.37 | 6.49 | 30.00 | 57.00 | 13.00 | 82 | 2.40 | 6.02 | 37.27 | 55.45 | 7.27 |
| 59 | 2.53 | 6.24 | 27.00 | 54.00 | 19.00 | 86 | 2.32 | 6.13 | 34.55 | 53.64 | 11.82 |
| 70 | 4.08 | 4.94 | 0.00 | 54.00 | 46.00 | 98 | 3.74 | 5.22 | 12.73 | 52.73 | 34.55 |
| 90 | 4.64 | 5.48 | 3.00 | 54.00 | 43.00 | 59 | 2.70 | 5.76 | 30.00 | 51.82 | 18.18 |
| 87 | 2.52 | 5.81 | 35.00 | 53.00 | 12.00 | 64 | 5.10 | 5.49 | 0.91 | 50.91 | 48.18 |
| 61 | 5.08 | 5.63 | 0.00 | 51.00 | 49.00 | 67 | 4.98 | 5.71 | 0.91 | 50.91 | 48.18 |
| 67 | 4.91 | 5.60 | 0.00 | 51.00 | 49.00 | 61 | 5.15 | 5.78 | 1.82 | 48.18 | 50.00 |
| 66 | 3.46 | 5.16 | 0.00 | 48.00 | 52.00 | 66 | 5.29 | 5.44 | 0.91 | 46.36 | 52.73 |
| 98 | 5.05 | 5.63 | 21.00 | 48.00 | 31.00 | 63 | 5.20 | 5.56 | 1.82 | 45.45 | 52.73 |
| 64 | 5.34 | 5.51 | 0.00 | 47.00 | 53.00 | | | | | | |
| 63 | 5.18 | 5.44 | 0.00 | 44.00 | 56.00 | | | | | | |

(*Continued.*)

(mean = 5.96, s.d. = 0.63) were rated as less negative than the images in the disgust-related group (mean = 2.79, s.d. = 0.89; $t_{197} = 24.58$, $p < 0.001$, $d = 4.12$) or the images in the fear-related group (mean = 3.21, s.d. = 1.03; $t_{197} = 17.72$, $p < 0.001$, $d = 3.24$). Moreover, the images in the disgust-related group were rated as more negative than the images in the fear-related group ($t_{197} = 2.58$, $p < 0.001$, $d = 0.44$). These findings suggest that the images in the disgust-related group were evaluated as most negative compared with the images in either colour or greyscaled conditions. This result supported Alarcão *et al.* [31], who also reported that the disgusting images were rated more negatively than the fearful images.

We conducted a one-way ANOVA with the image group (disgust-related, fear-related and neither groups) as independent variables and the mean arousal ratings for coloured images as the dependent variables. The results indicated that the main effect of image group was significant, $F_{2,197} = 358.66$, $p < 0.001$, $\eta_p^2 = 0.78$. *Post hoc* comparisons showed that the images in the neither group (mean = 3.70, s.d. = 0.46) were rated as less arousing than the images in the disgust-related group (mean = 5.65, s.d. = 0.74; $t_{197} = 21.18$, $p < 0.001$, $d = 3.16$) or the fear-related group (mean = 6.15, s.d. = 0.49; $t_{197} = 21.18$, $p < 0.001$, $d = 5.20$). Moreover, the images in the fear-related group were rated as more arousing than the images in the disgust-related group, $t_{197} = 4.41$, $p < 0.001$, $d = 0.80$. These findings suggest that the mean arousal ratings of image subgroups were different. The main effect of image group was significant for the mean arousal ratings of greyscaled images, $F_{2,197} = 270.68$, $p < 0.001$, $\eta_p^2 = 0.73$. A *post hoc* test of arousal ratings among image subgroups indicated that images in the neither group (mean = 3.45, s.d. = 0.51) were rated as less arousing than the images in the disgust-related group (mean = 5.50, s.d. = 0.94; $t_{197} = 18.92$, $p < 0.001$, $d = 2.71$) or in the fear-related group (mean = 5.99, s.d. = 0.44; $t_{197} = 19.47$, $p < 0.001$, $d = 5.34$). Moreover, the images in the fear-related group were rated as more arousing than the images in the disgust-related group, $t_{197} = 3.59$, $p < 0.001$, $d = 0.67$. These findings suggest that the mean arousal ratings of images in the fear-related group were the highest among the image groups under the colour and the greyscale conditions.

Figure 3 shows the relationship between valence and arousal ratings of images in each subgroup. The images in the disgust-related and fear-related groups in colour and greyscale conditions appear to be rated as more negative and more arousing than the images in the neither group. Moreover, the distribution of images in the disgust-related group appears to be more negative and less arousing than the distribution of images in the fear-related group.

| Cluster 3 in colour condition: Neither group | | | | | |
| --- | --- | --- | --- | --- | --- |
| Image ID | Valence | Arousal | Disgust ratio | Fear ratio | Neither ratio |
| 101 | 7.17 | 3.99 | 0.00 | 0.00 | 100.00 |
| 115 | 6.01 | 2.76 | 0.00 | 0.00 | 100.00 |
| 124 | 7.25 | 4.02 | 0.00 | 0.00 | 100.00 |
| 125 | 7.04 | 3.78 | 0.00 | 0.00 | 100.00 |
| 142 | 6.34 | 3.56 | 0.00 | 0.00 | 100.00 |
| 154 | 7.15 | 3.87 | 0.00 | 0.00 | 100.00 |
| 155 | 6.67 | 3.22 | 0.00 | 0.00 | 100.00 |
| 160 | 6.74 | 3.28 | 0.00 | 0.00 | 100.00 |
| 161 | 6.99 | 3.47 | 0.00 | 0.00 | 100.00 |
| 162 | 6.72 | 3.34 | 0.00 | 0.00 | 100.00 |
| 163 | 7.07 | 3.71 | 0.00 | 0.00 | 100.00 |
| 165 | 6.41 | 3.42 | 0.00 | 0.00 | 100.00 |
| 170 | 6.62 | 3.52 | 0.00 | 0.00 | 100.00 |
| 171 | 6.13 | 3.10 | 0.00 | 0.00 | 100.00 |
| 113 | 6.80 | 4.08 | 0.00 | 1.00 | 99.00 |
| 121 | 6.62 | 3.25 | 0.00 | 1.00 | 99.00 |
| 123 | 6.40 | 3.42 | 0.00 | 1.00 | 99.00 |
| 129 | 6.73 | 3.93 | 0.00 | 1.00 | 99.00 |
| 143 | 6.33 | 3.58 | 1.00 | 0.00 | 99.00 |
| 145 | 6.42 | 3.64 | 1.00 | 0.00 | 99.00 |
| 156 | 6.39 | 3.16 | 0.00 | 1.00 | 99.00 |
| 167 | 6.58 | 3.50 | 1.00 | 0.00 | 99.00 |
| 169 | 6.67 | 3.66 | 1.00 | 0.00 | 99.00 |
| 175 | 5.93 | 3.46 | 1.00 | 0.00 | 99.00 |
| 178 | 6.13 | 3.43 | 1.00 | 0.00 | 99.00 |
| 180 | 6.31 | 3.40 | 1.00 | 0.00 | 99.00 |
| 111 | 6.36 | 2.81 | 0.00 | 2.00 | 98.00 |
| 112 | 5.71 | 2.66 | 1.00 | 1.00 | 98.00 |
| 117 | 6.53 | 3.56 | 0.00 | 2.00 | 98.00 |
| 118 | 7.12 | 4.06 | 0.00 | 2.00 | 98.00 |
| 122 | 7.11 | 4.01 | 2.00 | 0.00 | 98.00 |
| 126 | 6.41 | 3.88 | 1.00 | 1.00 | 98.00 |
| 130 | 6.88 | 4.00 | 0.00 | 2.00 | 98.00 |
| 141 | 6.40 | 3.57 | 0.00 | 2.00 | 98.00 |
| 144 | 6.43 | 3.66 | 0.00 | 2.00 | 98.00 |
| 164 | 6.39 | 3.53 | 1.00 | 1.00 | 98.00 |
| 166 | 6.36 | 3.37 | 1.00 | 1.00 | 98.00 |
| 179 | 7.63 | 4.69 | 2.00 | 0.00 | 98.00 |
| 183 | 6.97 | 4.21 | 2.00 | 0.00 | 98.00 |
| 120 | 6.71 | 3.86 | 1.00 | 2.00 | 97.00 |
| 127 | 6.63 | 3.84 | 0.00 | 3.00 | 97.00 |
| 158 | 6.56 | 3.48 | 3.00 | 0.00 | 97.00 |
| 159 | 6.25 | 3.62 | 2.00 | 1.00 | 97.00 |
| 177 | 6.38 | 3.56 | 3.00 | 0.00 | 97.00 |
| 182 | 6.65 | 4.15 | 1.00 | 2.00 | 97.00 |
| 185 | 6.51 | 4.02 | 1.00 | 2.00 | 97.00 |
| 190 | 7.47 | 4.75 | 2.00 | 1.00 | 97.00 |
| 102 | 6.94 | 4.31 | 2.00 | 2.00 | 96.00 |
| 128 | 6.25 | 3.80 | 1.00 | 3.00 | 96.00 |
| 131 | 6.54 | 3.87 | 3.00 | 1.00 | 96.00 |
| 148 | 6.00 | 3.62 | 3.00 | 1.00 | 96.00 |
| 168 | 6.90 | 4.66 | 1.00 | 3.00 | 96.00 |
| 174 | 5.62 | 3.32 | 3.00 | 1.00 | 96.00 |
| 176 | 6.08 | 3.55 | 3.00 | 1.00 | 96.00 |
| 184 | 6.70 | 4.07 | 1.00 | 3.00 | 96.00 |
| 196 | 5.56 | 2.77 | 4.00 | 0.00 | 96.00 |
| 199 | 6.31 | 3.81 | 2.00 | 2.00 | 96.00 |
| 110 | 5.61 | 2.85 | 4.00 | 1.00 | 95.00 |
| 119 | 6.57 | 3.57 | 2.00 | 3.00 | 95.00 |
| 138 | 5.51 | 3.32 | 4.00 | 1.00 | 95.00 |
| 140 | 5.57 | 3.42 | 4.00 | 1.00 | 95.00 |
| 197 | 5.38 | 2.82 | 4.00 | 1.00 | 95.00 |
| 198 | 6.07 | 4.05 | 2.00 | 3.00 | 95.00 |
| 137 | 5.74 | 3.65 | 4.00 | 2.00 | 94.00 |
| 152 | 5.79 | 3.77 | 4.00 | 2.00 | 94.00 |
| 186 | 6.51 | 4.08 | 5.00 | 1.00 | 94.00 |
| 200 | 5.26 | 2.81 | 5.00 | 1.00 | 94.00 |
| 149 | 6.05 | 3.76 | 5.00 | 2.00 | 93.00 |
| 194 | 5.01 | 3.18 | 8.00 | 0.00 | 92.00 |
| 116 | 5.92 | 3.87 | 3.00 | 6.00 | 91.00 |
| 133 | 5.61 | 3.80 | 6.00 | 3.00 | 91.00 |
| 187 | 6.23 | 4.12 | 5.00 | 4.00 | 91.00 |
| 147 | 5.40 | 4.09 | 5.00 | 5.00 | 90.00 |
| 181 | 5.81 | 4.33 | 8.00 | 2.00 | 90.00 |
| 188 | 6.27 | 3.98 | 7.00 | 3.00 | 90.00 |
| 195 | 4.98 | 3.23 | 9.00 | 1.00 | 90.00 |
| 139 | 5.44 | 3.86 | 10.00 | 1.00 | 89.00 |
| 132 | 5.83 | 4.05 | 8.00 | 4.00 | 88.00 |
| 134 | 5.73 | 3.97 | 9.00 | 3.00 | 88.00 |
| 151 | 5.20 | 3.71 | 10.00 | 2.00 | 88.00 |
| 136 | 5.32 | 4.36 | 11.00 | 2.00 | 87.00 |
| 172 | 4.80 | 3.51 | 11.00 | 2.00 | 87.00 |
| 103 | 4.96 | 3.12 | 15.00 | 1.00 | 84.00 |
| 150 | 5.13 | 3.90 | 12.00 | 4.00 | 84.00 |
| 173 | 5.08 | 3.45 | 16.00 | 1.00 | 83.00 |
| 192 | 5.16 | 4.58 | 14.00 | 4.00 | 82.00 |
| 104 | 5.99 | 4.74 | 8.00 | 11.00 | 81.00 |
| 135 | 5.03 | 4.23 | 10.00 | 9.00 | 81.00 |
| 153 | 5.13 | 3.56 | 16.00 | 3.00 | 81.00 |
| 157 | 4.92 | 3.95 | 15.00 | 5.00 | 80.00 |
| 146 | 5.18 | 4.45 | 15.00 | 6.00 | 79.00 |
| 108 | 4.81 | 3.70 | 23.00 | 2.00 | 75.00 |
| 114 | 4.60 | 3.33 | 27.00 | 0.00 | 73.00 |

| Cluster 3 in greyscale condition: Neither group | | | | | |
| --- | --- | --- | --- | --- | --- |
| Image ID | Valence | Arousal | Disgust ratio | Fear ratio | Neither ratio |
| 111 | 5.79 | 2.23 | 0.00 | 0.00 | 100.00 |
| 115 | 5.60 | 2.22 | 0.00 | 0.00 | 100.00 |
| 163 | 6.85 | 3.12 | 0.00 | 0.00 | 100.00 |
| 160 | 6.50 | 3.04 | 0.91 | 0.00 | 99.09 |
| 184 | 7.15 | 4.01 | 0.91 | 0.00 | 99.09 |
| 125 | 6.39 | 3.28 | 1.82 | 0.00 | 98.18 |
| 130 | 6.92 | 3.98 | 0.91 | 0.91 | 98.18 |
| 143 | 6.28 | 3.66 | 0.91 | 0.91 | 98.18 |
| 155 | 6.48 | 2.77 | 0.91 | 0.91 | 98.18 |
| 167 | 6.44 | 3.17 | 1.82 | 0.00 | 98.18 |
| 179 | 7.55 | 4.43 | 0.91 | 0.91 | 98.18 |
| 199 | 6.67 | 4.30 | 1.82 | 0.00 | 98.18 |
| 118 | 6.56 | 3.71 | 1.82 | 0.91 | 97.27 |
| 124 | 6.85 | 3.95 | 0.91 | 1.82 | 97.27 |
| 144 | 6.38 | 3.43 | 0.91 | 1.82 | 97.27 |
| 145 | 6.32 | 3.56 | 0.91 | 1.82 | 97.27 |
| 161 | 6.32 | 2.85 | 2.73 | 0.00 | 97.27 |
| 164 | 6.40 | 3.19 | 2.73 | 0.00 | 97.27 |
| 171 | 5.94 | 2.77 | 2.73 | 0.00 | 97.27 |
| 173 | 5.34 | 2.82 | 2.73 | 0.00 | 97.27 |
| 182 | 6.60 | 3.94 | 2.73 | 0.00 | 97.27 |
| 183 | 7.12 | 4.54 | 0.91 | 1.82 | 97.27 |
| 101 | 7.12 | 3.99 | 0.91 | 2.73 | 96.36 |
| 123 | 5.75 | 3.12 | 3.64 | 0.00 | 96.36 |
| 129 | 6.70 | 3.81 | 1.82 | 1.82 | 96.36 |
| 141 | 6.30 | 3.43 | 2.73 | 0.91 | 96.36 |
| 148 | 6.15 | 3.66 | 2.73 | 0.91 | 96.36 |
| 149 | 6.08 | 3.55 | 1.82 | 1.82 | 96.36 |
| 162 | 6.00 | 2.87 | 2.73 | 0.91 | 96.36 |
| 166 | 6.33 | 3.11 | 3.64 | 0.00 | 96.36 |
| 185 | 6.71 | 3.97 | 2.73 | 0.91 | 96.36 |
| 190 | 7.26 | 4.34 | 0.91 | 2.73 | 96.36 |
| 122 | 6.38 | 3.32 | 4.55 | 0.00 | 95.45 |
| 137 | 5.86 | 3.15 | 3.64 | 0.91 | 95.45 |
| 142 | 6.24 | 3.32 | 1.82 | 2.73 | 95.45 |
| 151 | 5.51 | 3.43 | 2.73 | 1.82 | 95.45 |
| 165 | 6.35 | 3.26 | 4.55 | 0.00 | 95.45 |
| 169 | 6.45 | 3.31 | 3.64 | 0.91 | 95.45 |
| 178 | 5.37 | 2.95 | 4.55 | 0.00 | 95.45 |
| 121 | 6.44 | 3.31 | 3.64 | 1.82 | 94.55 |
| 127 | 6.26 | 3.38 | 4.55 | 0.91 | 94.55 |
| 128 | 6.23 | 3.73 | 2.73 | 2.73 | 94.55 |
| 131 | 6.12 | 3.62 | 1.82 | 3.64 | 94.55 |
| 152 | 5.75 | 3.31 | 3.64 | 1.82 | 94.55 |
| 158 | 6.44 | 3.25 | 4.55 | 0.91 | 94.55 |
| 159 | 5.93 | 3.09 | 3.64 | 1.82 | 94.55 |
| 180 | 5.36 | 2.93 | 5.45 | 0.00 | 94.55 |
| 196 | 5.47 | 2.94 | 4.55 | 0.91 | 94.55 |
| 200 | 5.31 | 2.86 | 5.45 | 0.00 | 94.55 |
| 102 | 6.58 | 4.21 | 1.82 | 4.55 | 93.64 |
| 138 | 5.35 | 3.05 | 5.45 | 0.91 | 93.64 |
| 154 | 6.45 | 3.64 | 5.45 | 0.91 | 93.64 |
| 156 | 5.92 | 2.92 | 6.36 | 0.00 | 93.64 |
| 168 | 6.31 | 4.25 | 1.82 | 4.55 | 93.64 |
| 170 | 6.25 | 3.15 | 5.45 | 0.91 | 93.64 |
| 194 | 5.05 | 3.02 | 5.45 | 0.91 | 93.64 |
| 197 | 5.22 | 3.04 | 6.36 | 0.00 | 93.64 |
| 119 | 6.17 | 3.47 | 5.45 | 1.82 | 92.73 |
| 133 | 5.59 | 3.35 | 7.27 | 0.00 | 92.73 |
| 176 | 5.98 | 3.50 | 6.36 | 0.91 | 92.73 |
| 186 | 6.70 | 4.40 | 6.36 | 0.91 | 92.73 |
| 120 | 6.02 | 3.57 | 5.45 | 2.73 | 91.82 |
| 134 | 5.60 | 3.45 | 6.36 | 1.82 | 91.82 |
| 172 | 5.19 | 2.65 | 7.27 | 0.91 | 91.82 |
| 198 | 6.08 | 4.34 | 4.55 | 3.64 | 91.82 |
| 103 | 5.16 | 3.03 | 8.18 | 0.91 | 90.91 |
| 113 | 6.19 | 3.85 | 3.64 | 5.45 | 90.91 |
| 175 | 5.56 | 3.23 | 8.18 | 0.91 | 90.91 |
| 117 | 5.64 | 3.35 | 8.18 | 1.82 | 90.00 |
| 140 | 5.45 | 3.22 | 8.18 | 1.82 | 90.00 |
| 187 | 6.24 | 4.45 | 8.18 | 1.82 | 90.00 |
| 188 | 6.23 | 4.25 | 8.18 | 1.82 | 90.00 |
| 112 | 5.05 | 2.61 | 8.18 | 2.73 | 89.09 |
| 150 | 5.40 | 3.43 | 7.27 | 3.64 | 89.09 |
| 181 | 5.55 | 3.87 | 6.36 | 4.55 | 89.09 |
| 132 | 5.55 | 3.42 | 11.82 | 0.00 | 88.18 |
| 147 | 5.65 | 3.94 | 4.55 | 7.27 | 88.18 |
| 136 | 5.30 | 3.43 | 10.91 | 1.82 | 87.27 |
| 139 | 5.48 | 3.53 | 9.09 | 3.64 | 87.27 |
| 116 | 5.28 | 3.25 | 10.91 | 2.73 | 86.36 |
| 104 | 5.66 | 4.53 | 5.45 | 10.00 | 84.55 |
| 107 | 4.72 | 3.19 | 10.91 | 4.55 | 84.55 |
| 126 | 5.39 | 3.28 | 13.64 | 1.82 | 84.55 |
| 157 | 5.18 | 3.65 | 10.91 | 5.45 | 83.64 |
| 177 | 5.50 | 3.73 | 14.55 | 1.82 | 83.64 |
| 110 | 4.84 | 2.77 | 14.55 | 2.73 | 82.73 |
| 135 | 5.12 | 4.02 | 7.27 | 10.00 | 82.73 |
| 174 | 4.98 | 3.07 | 16.36 | 0.91 | 82.73 |
| 153 | 5.26 | 3.60 | 12.73 | 6.36 | 80.91 |
| 146 | 4.95 | 4.26 | 9.09 | 10.91 | 80.00 |
| 195 | 4.65 | 3.39 | 20.00 | 0.91 | 79.09 |

We conducted a correlation analysis between valence and arousal ratings of the images in disgust-related and fear-related groups to examine differences in valence and arousal ratings between the groups. As can be seen in figure 3, the mean valence ratings of the images were negatively correlated with the mean arousal ratings of the images in the fear- and disgust-related groups in both

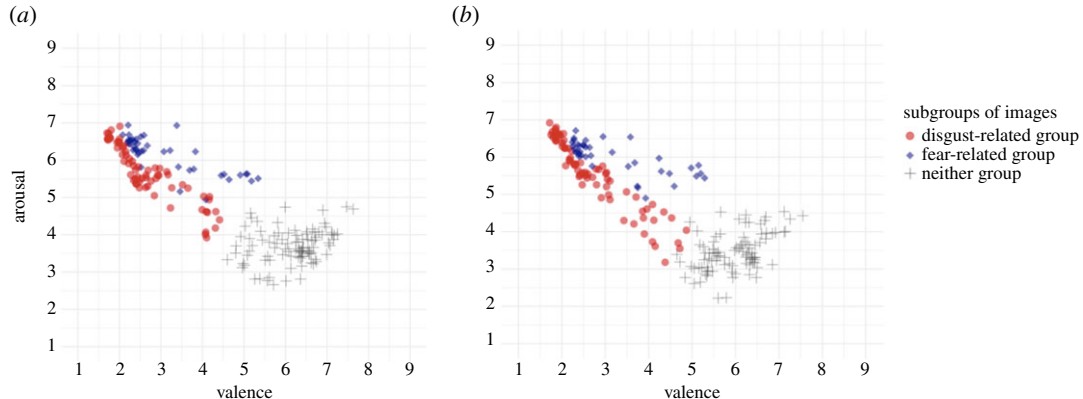

**Figure 3.** Relationship between valence and arousal ratings for images in each subgroup (disgust-related, fear-related, or neither group). (*a*) shows the colour condition. (*b*) shows the greyscale condition.

**Table 2.** Numbers of images per subgroups.

| mode of presentation | | name of clusters | greyscale condition | | | |
|---|---|---|---|---|---|---|
| | | | cluster 1 | cluster 2 | cluster 3 | |
| | | | disgust-related group | fear-related group | neither group | total |
| colour | cluster 1 | disgust-related group | 65 | 0 | 1 | 66 |
| condition | cluster 2 | fear-related group | 2 | 39 | 0 | 41 |
| | cluster 3 | neither group | 3 | 0 | 90 | 93 |
| | | total | 70 | 39 | 91 | 200 |

presentation modes (fear-related group in colour condition: $r = -0.78$ [$-0.88$; $-0.63$], $t_{39} = -7.89$, $p < 0.001$; disgust-related group in colour condition: $r = -0.90$ ($-0.94$; $-0.84$), $t_{64} = -16.58$, $p < 0.01$; fear-related group in greyscale condition: $r = -0.72$ ($-0.84$; $-0.52$), $t_{37} = -6.28$, $p < 0.001$; disgust-related group in greyscale condition: $r = -0.94$ ($-0.97$; $-0.91$), $t_{68} = -23.67$, $p < 0.001$). Moreover, the correlations between valence and arousal ratings in the disgust-related group were stronger than those in the fear-related group in both presentation modes (colour condition, $z = 2.04$, $p = 0.04$; greyscale condition, $z = 4.22$, $p < 0.001$), suggesting that the strength of relationship between valence and arousal might be different between discrete emotions such as disgust and fear.

The results of Experiment 1 demonstrated the probability of each image being categorized as disgusting or fearful. However, the extent to which participants felt disgust or fear for each image was unclear. Therefore, in Experiment 2, we examined the level of disgust and fear felt for each image by asking participants to rate the degree of disgust or fear they felt in response to each image.

# 3. Experiment 2

## 3.1. Methods

We recruited 423 workers (134 females and 289 males, mean age = 46.38 years, age range = 18–78 years) through another crowdsourcing service (Yahoo! Crowdsourcing). They participated in Experiment 2 and evaluated the images. Similar to Experiment 1, most of the participants were Japanese and/or understood Japanese. All participants gave their informed consent online prior to participating in the study. Ethical approval for this study was obtained from the Ethical Review Committee for Research Involving Human Subjects at Waseda University.

The procedure of Experiment 2 was identical to that of Experiment 1, except for the following. The participants were randomly assigned to the fear-rating group or the disgust-rating group. Participants

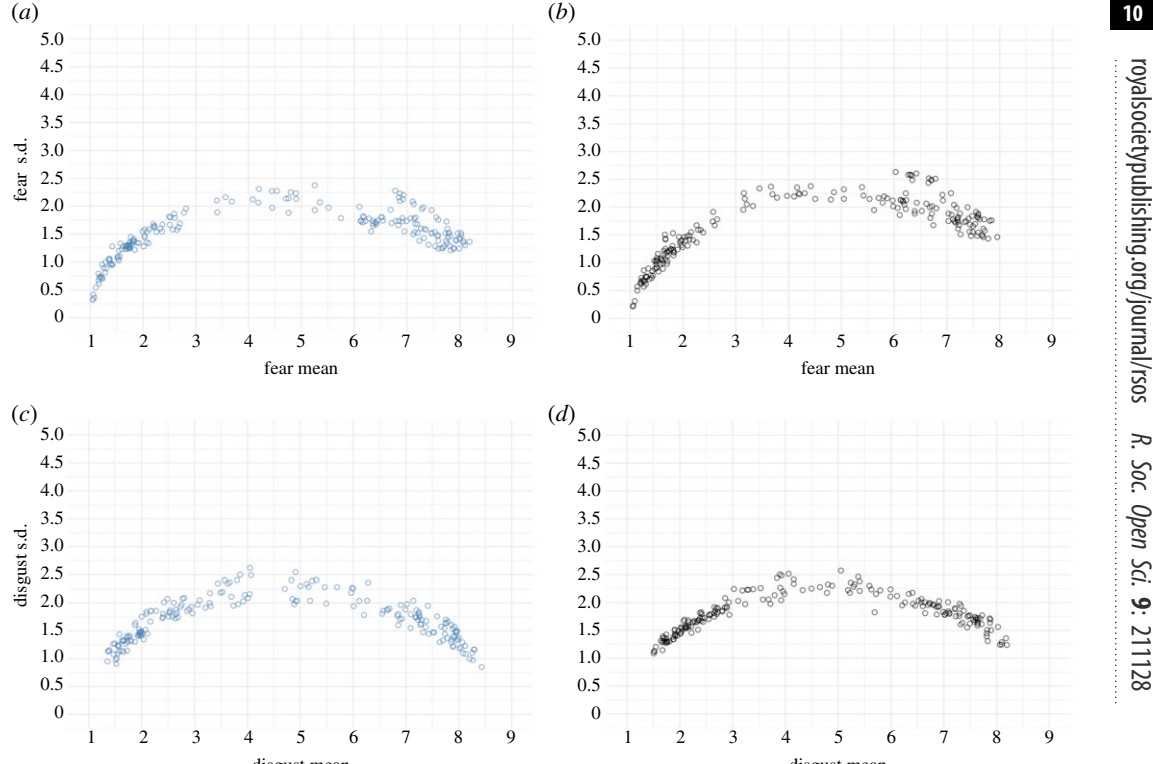

**Figure 4.** Relationships between the mean values and standard deviations of fear (*a,b*) and disgust ratings (*c,d*). (*a,c*) show the colour condition, and (*b,d*) display the greyscale condition.

in each group were further assigned to a colour image group (fear-rating and colour condition $N = 103$; disgust-rating and colour condition $N = 102$) and a greyscale image group (fear-rating and greyscale condition $N = 105$; disgust-rating and greyscale condition $N = 113$). They were instructed to rate each image on disgust or fear dimensions (appendix B). Disgust was assessed by a 9-point scale ranging from 1 (*Low disgust*) to 9 (*High disgust*), and fear was also assessed by a 9-point scale ranging from 1 (*Low fear*) to 9 (*High fear*).

## 3.2. Results and discussion

### 3.2.1. Univariate distributions

Figure 4 shows relationships between fear and disgust ratings' means and standard deviations. The fear ratings for the images ranged from 1.05 to 8.18 (*mean* = 4.54) in the colour condition and from 1.05 to 7.95 (*mean* = 4.28) in the greyscale condition. Disgust ratings ranged from 1.35 to 8.43 (*mean* = 4.64) in colour and 1.50 to 8.19 (*mean* = 4.54) in the greyscale condition.

The face validity of the fear and disgust ratings for the images was assessed by probing the most fearful, most disgusting, least fearful and least disgusting images for each presentation mode. In the colour condition, the most fearful image was a snake (image no. 76; mean fear = 8.18, s.d. = 1.36), and the most disgusting image was a millipede (image no. 13; mean disgust = 8.43, s.d. = 0.85), whereas the least fearful (image no. 160, 163; mean fear = 1.05, s.d. = 0.32), and least disgusting image (image no. 161; mean disgust = 1.35, s.d. = 0.95) was an orange. In the greyscale condition, the most fearful (image no. 16, mean = 7.95, s.d. = 1.46) and the most disgusting image (image no. 16, mean = 8.19, s.d. = 1.24) was a centipede, whereas the least fearful image was an orange (image no. 163; mean fear = 1.05, s.d. = 0.21), and the least disgusting image was an apple (image no. 160; mean disgust = 1.50, s.d. = 1.09).

### 3.2.2. Relationship between disgust and fear ratings

We examined the relationship between disgust and fear ratings for images under each presentation mode. The results showed that the mean fear ratings of the images were positively correlated with the

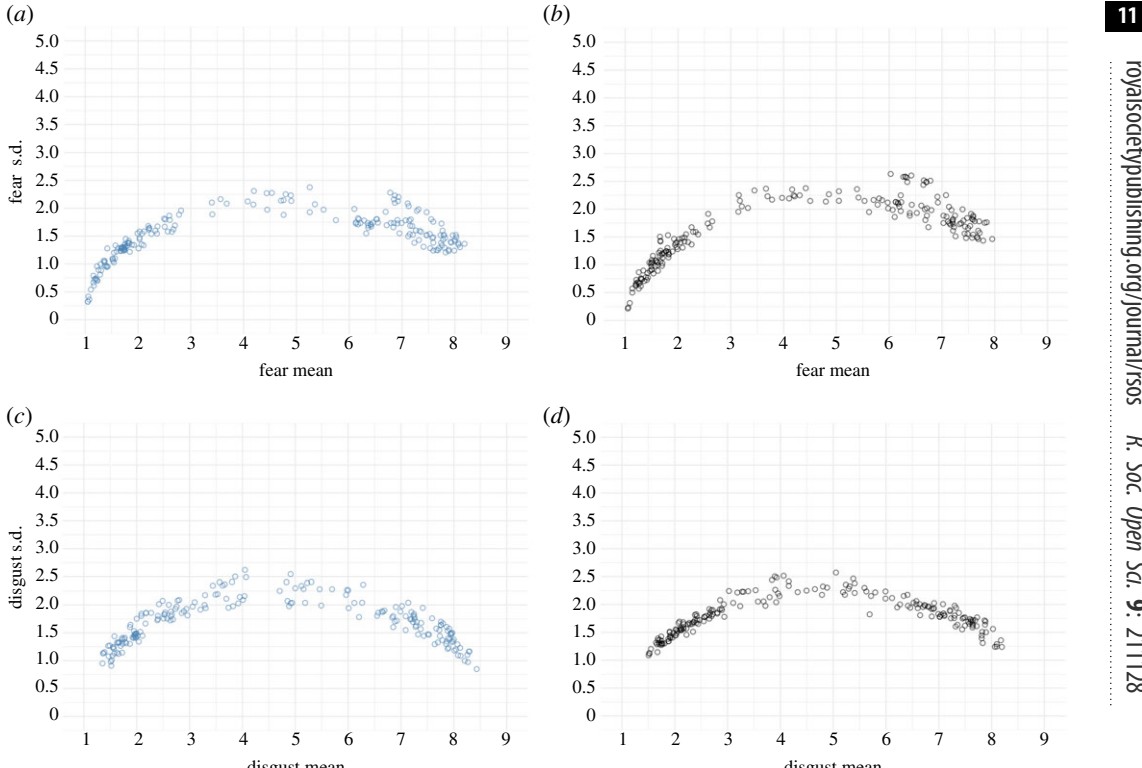

**Figure 5.** Relationship between fear and disgust ratings by gender. (*a*) shows the colour condition, and (*b*) displays the greyscale condition.

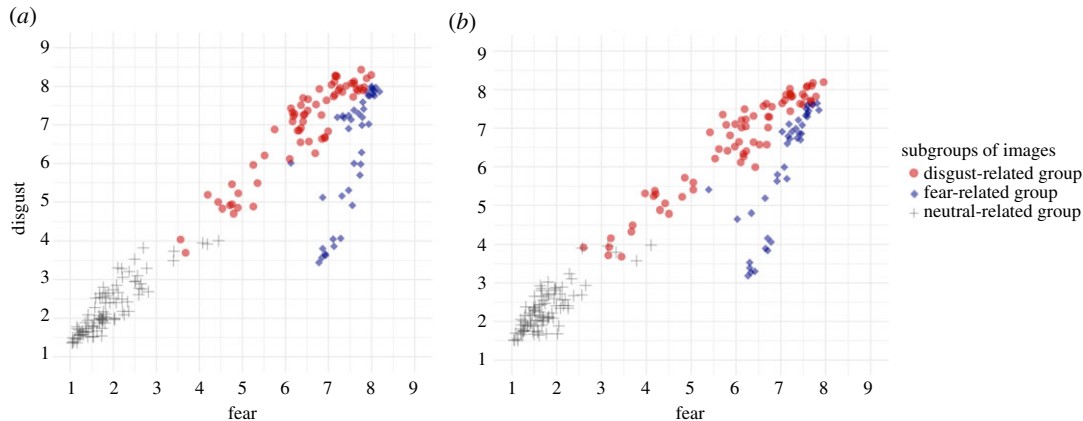

subgroups of images
● disgust-related group
♦ fear-related group
+ neutral-related group

**Figure 6.** Relationship between fear and disgust ratings of images in each subgroup (disgust-related, fear-related or neither group). (*a*) shows the colour condition, and (*b*) shows the greyscale condition. The colour and shape represent the subgroups of images in Experiment 1.

mean disgust ratings under colour and greyscale conditions (colour condition: $r = 0.94$ [0.92; 0.95], $t_{198} = 37.75$, $p < 0.001$; greyscale condition: $r = 0.94$ [0.92; 0.95], $t_{198} = 37.91$, $p < 0.001$), suggesting that in general, the more disgusting was an OBNIS image, the more fearful it was.

### 3.2.2.1. Gender differences

We also found high positive correlations between disgust and fear ratings in both genders (figure 5; women in the colour condition: $r = 0.95$ (0.94; 0.96), $t_{198} = 44.21$, $p < 0.001$; men in the colour condition: $r = 0.93$ (0.90; 0.94), $t_{198} = 34.46$, $p < 0.001$; women in the greyscale condition: $r = 0.95$ (0.94; 0.96), $t_{198} = 43.84$, $p < 0.001$; men in the greyscale condition: $r = 0.93$ (0.91; 0.95), $t_{198} = 34.96$, $p < 0.001$). The

correlation between fear and disgust ratings was stronger in women than men under both presentation modes (colour condition: $z = -2.32$, $p = 0.02$; greyscale condition: $z = -2.11$, $p = 0.03$).

### 3.2.2.2. Cluster differences

Figure 6 shows relationships between fear and disgust in the subgroups categorized by the cluster analysis of Experiment 1 (i.e. disgust-related, fear-related and neither groups). We tested the relationships between fear and disgust in the disgust-related and fear-related groups. A correlation analysis indicated that the mean fear ratings of the images were positively correlated with the mean disgust ratings of the images in all image subgroups (disgust-related group in colour condition: $r = 0.93$ (0.89; 0.96), $t_{64} = 20.28$, $p < 0.001$; fear-related group in colour condition: $r = 0.76$ (0.59; 0.87), $t_{39} = 7.36$, $p < 0.001$; neither group in colour condition: $r = 0.88$ (0.82; 0.92), $t_{91} = 17.56$, $p < 0.001$; disgust-related group in greyscale condition: $r = 0.95$ (0.93; 0.97), $t_{68} = 26.12$, $p < 0.001$; fear-related group in greyscale condition: $r = 0.83$ [0.70; 0.91], $t_{37} = 9.22$, $p < 0.001$; neither group in greyscale condition: $r = 0.81$ (0.72; 0.87), $t_{89} = 12.95$, $p < 0.001$). Moreover, the correlation between fear and disgust ratings was stronger in the disgust-related than the fear-related group under both presentation modes (colour condition: $z = 3.20$, $p < 0.01$; greyscale condition: $z = 3.23$, $p < 0.01$). A visual inspection indicated that the fear-related group contained more images that induced less disgust and more fear than the disgust-related group. These images might have caused the weaker correlation between fear and disgust ratings in the fear-related compared with the disgust-related group. Less disgusting images in the fear-related group mainly included images of carnivorous mammals, including tigers and lions, whereas the more disgusting images in the fear-related group mainly included images of reptiles and insects such as snakes and wasps. The possible differences in the relationship between disgust and fear ratings among different animal species, even among images that are frequently categorized as 'fearful', need to be further investigated in the future.

## 4. General discussion

We collected 200 images to provide a comprehensive database for investigating visually elicited negative emotions, which we call the 'Open Biological Negative Image Set.' The OBNIS includes valence, arousal, disgust and fear levels, and the emotional category (disgust, fear or neither) of each image. The images and their ratings are fully available on the Internet (https://osf.io/pfrx4/?view_only=911b1be722074ad4aab87791cb8a72f5). The images in the OBNIS, similar to those in the IAPS, show that valence and arousal variability ratings form a nearly uniform distribution. IAPS images show a boomerang-shaped relationship between the mean valence and arousal ratings such that extremely positive and negative images had the highest arousal scores. Conversely, the OBNIS showed a linear relationship between the mean valence and arousal ratings. This difference could be because the OBNIS mainly contains emotionally negative images. As a result, the variation of arousal ratings on positive valence might be small. Furthermore, the images in OBNIS were relatively well divided into three groups (disgust-related, fear-related and neither group), regardless of the presentation mode. However, in specific images, especially in the disgust-related group, the possibility of 'fear' was as high as the probability of 'disgust'. Thus, it is possible that disgust and fear tend to be confused, especially when viewing images associated with disgusting objects. Indeed, the images classified as the disgust-related group in Experiment 2 showed a strong correlation between disgust and fear ratings, indicating that even when images are classified as disgusting, people sometimes feel fearful and disgusted.

In conclusion, the OBNIS provides ratings on valence, arousal, disgust, fear and emotional categories, including 'disgust', 'fear' or 'neither' for 200 images. Moreover, the OBNIS provides colour and greyscale versions of the images. We believe that the OBNIS will be a useful image resource for studying visually evoked negative emotions such as disgust and fear in affective research.

Data accessibility. The data and materials for our experiment are available at https://osf.io/pfrx4/?view_only=911b1be722074ad4aab87791cb8a72f5. This experiment was not preregistered.

Authors' contributions. R.S. designed the study, conducted the online experiments and analyses; K.W. designed and coordinated the study. All authors contributed to the writing of the final manuscript. All authors gave final approval for publication and agreed to be held accountable for the work performed therein.

Competing interests. We have no conflicts of interest to report regarding the finding of this study.

Funding. This work was supported by JSPS KAKENHI Grant no. JP20J00838; KAKENHI Grant no. 17H06344, 17H00753; JST-CREST JPMJCR16E1; JST-MIRAI program Grant no. JPMJMI20D8; JST-Moonshot Research and Development Grant no. JPMJMS2012.

# Appendix A—Instructions

## [Question 1—valence]

How strong is your negative or positive emotion when you see this image? Please choose a number ranging from 1 (*Very negative*) to 9 (*Very positive*) that reflects your feelings most closely.

 * Please make your evaluation based on your own feelings, rather than on how people generally feel.

## [Question 2—arousal]

How aroused do you feel when you see the image? Please choose a number ranging from 1 (*Low arousal*) to 9 (*High arousal*) that reflects your feelings most closely.

 * 'How aroused do you feel?' means how much excitement, surprise, agitation or pounding of the heart do you feel? 1 (Low arousal) means very calm, and 9 (High arousal) means very excited with higher numbers expressing increased arousal. The degree of arousal is independent of the response to Question 1. Therefore, the degree of arousal might be high (close to 9) or lower (close to 1) despite the images' emotional valence.

## [Question 3—categorization]

Which emotion do you feel, 'disgust' or 'fear' when you see this image? Please select 'neither', if you do not feel either one of these two emotions.

 * 'Disgust' describes repugnance, irritability and a feeling of heartburn. On the other hand, 'Fear' indicates a fear of imminent danger, distress, or anxiety. For example, people expect to feel disgusted when they see an image of dirty toilets, and people are expected to feel fearful when they see an image of unexploded shells. If you do not feel 'disgust' or 'fear', please select 'neither'. Please note that you should evaluate fear or disgust based on how you feel, rather than on how people generally feel.

# Appendix B—Instructions

## [Question—disgust]

How strong is your disgust when you see this image? Please choose a number between 1 (*Low disgust*) to 9 (*High disgust*) that most closely reflects your feelings.

 * 'Disgust' describes repugnance, irritability and a feeling of heartburn. Please make your evaluation based on your own feelings rather than how people generally feel.

## [Question—fear]

How strong is your fear when you see this image? Please choose a number between 1 (*Low disgust*) to 9 (*High disgust*) that most closely reflects your feelings.

 * 'Fear' describes a feeling of imminent danger, distress or anxiety. Please make your evaluation based on your own feelings rather than how people generally feel.

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
