## [Peer Review File · Royal Society Open Science]

Review History

RSOS-211128.R0 (Original submission)

Review form: Reviewer 1 (David S. March)

Is the manuscript scientifically sound in its present form?

Yes

Are the interpretations and conclusions justified by the results?

No

Is the language acceptable?

Yes

Do you have any ethical concerns with this paper?

No

Have you any concerns about statistical analyses in this paper?

Yes

Recommendation?

Accept with minor revision (please list in comments)

Comments to the Author(s)

This is a review of the manuscript titled, "Open Biological Negative Image Set." There are a lot of image sets currently available, so justifying a new one is paramount to its utility. The current set fills a nice niche. I don't know of another set where all the images have been de-contextualized as cleanly. I imagine the work it took to produce these images was no small feat and I commend the authors for the high quality of their images. I think the set will be useful, though there are certain drawbacks to the way in which they piloted the images and the way they report their findings that could use clarification.

I'm unclear why the authors chose to use a dichotomous yes this is fear vs disgust vs neither label instead of using a scale for each. In as much as disgust images can also evoke fear, it's important to know which disgust images are low threat and which are high threat. Indeed, my earlier work (March et al., 2017) found unique responses to threat vs disgust, and when I piloted images for that work, it became clear that isolating threat from disgust was important were I to make distinctions between responses to threat vs response to disgust. In as much as threat is confounded with disgust in the current work, it's hard to justify the use of the author's disgust images as purely representing disgust and not threat. Here participants are forced to choose between "disgusting" or "fearful". Had the authors used a scale for each, it would likely have been clear that some images are disgusting AND fearful, while others are high on one but low on the other. Forcing a category onto the image may imply that one is more disgust than fearful (or vice versa), but it certainly does not imply that the image is disgust and not fearful. This is an important distinction that the authors completely ignore. If possible, I'd highly recommend they collect this data as it would make for a much more useful image set.

This decision is made more confusing by the fact that the authors collected valence ratings on a continuous scale.

In their discussion of the "relationship between valence and arousal" the authors gloss over the distinction between positive and negative valence. There is a very high negative correlation between arousal and valence, and looking at Figure 2 seems to imply that arousal is highest among negative images. In fact, it looks like there is no correlation between valence and arousal for positive images.

The authors also state that "the relationship between valence and arousal is stronger in women than men"; I'd again like to know whether this is true for both positive and negative images. It looks like only for negative images. In the greyscale, there's an obvious group of neutrally valenced images that women but not men find highly arousing only in the grey and not in the color. This is very strange. What are these images and why would women differ? Indeed, it's unclear why any neutrally valenced image would be highly arousing.

Figure 3 (which is really a table) is great and all, but would really benefit from being split onto multiple pages. The text is tiny. In addition, it should have the valence and arousal ratings of each image in the table.

On page 17, the authors mention a finding that "negative feelings experiencing when viewing disgusting images are stronger than when viewing fear images." My earlier work would disagree. In that piloting, I found that many threat images (e.g., tiger) also evoke positivity (they are cool, beautiful creatures). So it's not that people feel more negative from disgust than threat, it's just that they feel less positive. Indeed, this position is supported by the author's reporting that arousal was stronger for the fear than disgust related group.

I partially disagree with the statement on page 18 that characterizes fear as "high arousal" and disgust as "low arousal". There are 25 more images in the disgust than fear set in the current work. Looking at Figure 4, at the high end of negative valence, the disgust vs fear correlation with arousal likely does not differ. It's only when you get to the less negative or even slightly positive

images that the correlation weakens. This implies that as the author's stated, threat stimuli are high arousal, but this certainly doesn't imply that disgust are low arousal. Looking again to Figure 4, there are plenty of high arousal disgust stimuli.

On page 21, the authors state that "highly arousing images tended to be consistently categorized as "fearful regardless of the valence rating. Recall my earlier point about certain threat stimuli being both positive and threat. I would suggest then that their statement is backward. It should instead read that "in this image set, high threat images tended to also be high arousal."

P21. In their mention of the idea "that the response patterns for disgust and fear have distinct physiological, behavioral, neural, and cognitive characteristics", I suggest they see the below referenced paper for more explanation.

They end the paper by focusing strongly on the idea that diffuse arousal leads to fear vs threat and base this on the previously discussed correlation. This completely missed the mark, in my mind. They really don't need to focus on interpreting their data, but simply presenting it. They're putting a lot of stock into people interpreting their arousal when classifying these images, and their data just don't seem to support that position. It really just seems like a cherry-picked finding. But when you look at the figures, it doesn't hold up.

All in all, I like the paper and the image set. The authors go a bit beyond where they need to when justifying the utility of the set, and make some claims that just aren't strongly supported by their data. I'd suggest constraining the unnecessary interpretation and just present the patterns. I'd also suggest, if possible, to collect rating data of threat and disgust, separately, instead of the threat vs disgust category.

Work cited:

March, D. S., Gaertner, L., & Olson, M. A. (2017). In harm's way: On preferential response to threatening stimuli. *Personality and Social Psychology Bulletin*, 43(11), 1519-1529.

Decision letter (RSOS-211128.R0)

Dear Dr Shirai

The Editors assigned to your paper RSOS-211128 "Open Biological Negative Image Set (OBNIS)" have now received comments from reviewers and would like you to revise the paper in accordance with the reviewer comments and any comments from the Editors. Please note this decision does not guarantee eventual acceptance.

Please submit your revised manuscript and required files (see below) no later than 21 days from today's (ie 28-Sep-2021) date. Note: the ScholarOne system will 'lock' if submission of the revision is attempted 21 or more days after the deadline. If you do not think you will be able to meet this deadline please contact the editorial office immediately.

on behalf of Dr Gina Grimshaw (Associate Editor) and Essi Viding (Subject Editor)
openscience@royalsociety.org

Associate Editor Comments to Author (Dr Gina Grimshaw):

Comments to the Author:

Dear Dr Shirai,

Thank you for submitting your manuscript to Royal Society Open Science. I have now received one expert review and had hoped for another, but we have been unable to secure a second review. However, I have read the manuscript carefully myself, and am comfortable making a recommendation based on this review and my own reading. I agree with the reviewer that your image set would be a valuable resource for emotion researchers. Despite the fact that a large number of emotional image sets are available, your images are particularly well prepared, and provide great range within the fear and disgust categories. However, I also agree with the reviewer that the manuscript could benefit from some revision and (ideally) an alternative set of ratings. I have classed this as a major revision because of the recommendation to collect more data, but the revisions themselves will be relatively minor. I will not repeat the reviewer's concerns here, but I will emphasise those I consider to be particularly important to address.

1. I agree with the reviewer that independent ratings of fear and disgust, ideally provided by different participants, would be more valuable than forcing raters to choose fear or disgust. Given that ratings can be collected online, this additional data should not be too onerous to collect, and I think would greatly extend the value of the image set.

2. I also agree that you should focus on descriptive statistics that characterise the image set, and not use the data to draw inferences about the nature of fear or disgust, their relative intensities, or their relationships with arousal. Since you cannot be sure you have drawn comprehensively from the fear and disgust categories, nor that fear and disgust are equally well conveyed through static 2-d images, any conclusions about fear and disgust per se would be a consequence of the specific images you have created. In my mind, a descriptive focus on the images does not diminish the value of the manuscript or image set at all.

The reviewer also makes a number of recommendations that should improve the impact of your work, and I encourage you to give them serious consideration. In addition, I'd like to see you add any cultural or geographic demographic information you may have on the participants who provided the ratings, given that emotion ratings are often not universal.

Kind regards
Gina Grimshaw

Reviewer comments to Author:

Reviewer: 1

Comments to the Author(s)

This is a review of the manuscript titled, "Open Biological Negative Image Set." There are a lot of image sets currently available, so justifying a new one is paramount to its utility. The current set fills a nice niche. I don't know of another set where all the images have been de-contextualized as cleanly. I imagine the work it took to produce these images was no small feat and I commend the authors for the high quality of their images. I think the set will be useful, though there are certain drawbacks to the way in which they piloted the images and the way they report their findings that could use clarification.

I'm unclear why the authors chose to use a dichotomous yes this is fear vs disgust vs neither label instead of using a scale for each. In as much as disgust images can also evoke fear, it's important to know which disgust images are low threat and which are high threat. Indeed, my earlier work (March et al., 2017) found unique responses to threat vs disgust, and when I piloted images for that work, it became clear that isolating threat from disgust was important were I to make distinctions between responses to threat vs response to disgust. In as much as threat is confounded with disgust in the current work, it's hard to justify the use of the author's disgust images as purely representing disgust and not threat. Here participants are forced to choose between "disgusting" or "fearful". Had the authors used a scale for each, it would likely have been clear that some images are disgusting AND fearful, while others are high on one but low on the other. Forcing a category onto the image may imply that one is more disgust than fearful (or vice versa), but it certainly does not imply that the image is disgust and not fearful. This is an important distinction that the authors completely ignore. If possible, I'd highly recommend they collect this data as it would make for a much more useful image set.

This decision is made more confusing by the fact that the authors collected valence ratings on a continuous scale.

In their discussion of the "relationship between valence and arousal" the authors gloss over the distinction between positive and negative valence. There is a very high negative correlation between arousal and valence, and looking at Figure 2 seems to imply that arousal is highest among negative images. In fact, it looks like there is no correlation between valence and arousal for positive images.

The authors also state that "the relationship between valence and arousal is stronger in women than men"; I'd again like to know whether this is true for both positive and negative images. It looks like only for negative images. In the greyscale, there's an obvious group of neutrally valenced images that women but not men find highly arousing only in the grey and not in the color. This is very strange. What are these images and why would women differ? Indeed, it's unclear why any neutrally valenced image would be highly arousing.

Figure 3 (which is really a table) is great and all, but would really benefit from being split onto multiple pages. The text is tiny. In addition, it should have the valence and arousal ratings of each image in the table.

On page 17, the authors mention a finding that "negative feelings experienced when viewing disgusting images are stronger than when viewing fear images." My earlier work would

disagree. In that piloting, I found that many threat images (e.g., tiger) also evoke positivity (they are cool, beautiful creatures). So it's not that people feel more negative from disgust than threat, it's just that they feel less positive. Indeed, this position is supported by the author's reporting that arousal was stronger for the fear than disgust related group.

I partially disagree with the statement on page 18 that characterizes fear as "high arousal" and disgust as "low arousal". There are 25 more images in the disgust than fear set in the current work. Looking at Figure 4, at the high end of negative valence, the disgust vs fear correlation with arousal likely does not differ. It's only when you get to the less negative or even slightly positive images that the correlation weakens. This implies that as the author's stated, threat stimuli are high arousal, but this certainly doesn't imply that disgust are low arousal. Looking again to Figure 4, there are plenty of high arousal disgust stimuli.

On page 21, the authors state that "highly arousing images tended to be consistently categorized as "fearful regardless of the valence rating. Recall my earlier point about certain threat stimuli being both positive and threat. I would suggest then that their statement is backward. It should instead read that "in this image set, high threat images tended to also be high arousal."

P21. In their mention of the idea "that the response patterns for disgust and fear have distinct physiological, behavioral, neural, and cognitive characteristics", I suggest they see the below referenced paper for more explanation.

They end the paper by focusing strongly on the idea that diffuse arousal leads to fear vs threat and base this on the previously discussed correlation. This completely missed the mark, in my mind. They really don't need to focus on interpreting their data, but simply presenting it. They're putting a lot of stock into people interpreting their arousal when classifying these images, and their data just don't seem to support that position. It really just seems like a cherry-picked finding. But when you look at the figures, it doesn't hold up.

All in all, I like the paper and the image set. The authors go a bit beyond where they need to when justifying the utility of the set, and make some claims that just aren't strongly supported by their data. I'd suggest constraining the unnecessary interpretation and just present the patterns. I'd also suggest, if possible, to collect rating data of threat and disgust, separately, instead of the threat vs disgust category.

Work cited:

March, D. S., Gaertner, L., & Olson, M. A. (2017). In harm's way: On preferential response to threatening stimuli. *Personality and Social Psychology Bulletin*, 43(11), 1519-1529.

===PREPARING YOUR MANUSCRIPT===

Please ensure that you include an acknowledgements' section before your reference list/bibliography. This should acknowledge anyone who assisted with your work, but does not

qualify as an author per the guidelines at <https://royalsociety.org/journals/ethics-policies/openness/>.

===PREPARING YOUR REVISION IN SCHOLARONE===

- Ensure that your data access statement meets the requirements at <https://royalsociety.org/journals/authors/author-guidelines/#data>. You should ensure that you cite the dataset in your reference list. If you have deposited data etc in the Dryad repository, please include both the 'For publication' link and 'For review' link at this stage.
- If you are requesting an article processing charge waiver, you must select the relevant waiver option (if requesting a discretionary waiver, the form should have been uploaded at Step 3 'File upload' above).
- If you have uploaded ESM files, please ensure you follow the guidance at <https://royalsociety.org/journals/authors/author-guidelines/#supplementary-material> to include a suitable title and informative caption. An example of appropriate titling and captioning may be found at [https://figshare.com/articles/Table_S2_from_Is_there_a_trade-off_between_peak_performance_and_performance_breadth_across_temperatures_for_aerobic_sc ope_in_teleost_fishes_/3843624](https://figshare.com/articles/Table_S2_from_Is_there_a_trade-off_between_peak_performance_and_performance_breadth_across_temperatures_for_aerobic_scope_in_teleost_fishes_/3843624).

Author's Response to Decision Letter for (RSOS-211128.R0)

See Appendix A.

Decision letter (RSOS-211128.R1)

Dear Dr Shirai,

It is a pleasure to accept your manuscript entitled "Open Biological Negative Image Set (OBNIS)" in its current form for publication in Royal Society Open Science. The comments of the reviewer(s) who reviewed your manuscript are included at the foot of this letter.

The proof of your paper will be available for review using the Royal Society online proofing system and you will receive details of how to access this in the near future from our production office (openscience_proofs@royalsociety.org). We aim to maintain rapid times to publication after

acceptance of your manuscript and we would ask you to please contact both the production office and editorial office if you are likely to be away from e-mail contact to minimise delays to publication. If you are going to be away, please nominate a co-author (if available) to manage the proofing process, and ensure they are copied into your email to the journal.

on behalf of Dr Gina Grimshaw (Associate Editor) and Essi Viding (Subject Editor)
openscience@royalsociety.org

Associate Editor Comments to Author (Dr Gina Grimshaw):

Associate Editor

Comments to the Author:

Thank you for submitting this revised manuscript. I have read the manuscript carefully myself, and find that I do not need to send it back to reviewers. I think you have addressed all concerns, and particularly appreciate the effort in collecting new fear and disgust ratings, which I think are a valuable addition. I look forward to seeing this work published.

Follow Royal Society Publishing on Twitter: [@RSocPublishing](https://twitter.com/RSocPublishing)

Appendix A

November 1st, 2021

Reference: RSOS-211128

We highly appreciate you for taking the time to make valuable comments on the above-entitled manuscript despite your busy schedule. We have revised the manuscript according to the comments. We have attached our responses to editor and reviewer's comments and indicated the changes we made in response to each comment. We hope that the revised version of the manuscript is suitable for publication. However, we would be happy to clarify your concerns and further revise the manuscript if you have any further questions or comments.

RESPONSE TO ASSOCIATE EDITOR:

Comment 1: I agree with the reviewer that independent ratings of fear and disgust, ideally provided by different participants, would be more valuable than forcing raters to choose fear or disgust. Given that ratings can be collected online, this additional data should not be too onerous to collect, and I think would greatly extend the value of the image set.

Response: Thank you for this comment. We conducted an additional survey and included the data of the independent ratings of fear and disgust according to your comment.

Comment 2: I also agree that you should focus on descriptive statistics that characterise the image set, and not use the data to draw inferences about the nature of fear or disgust, their relative intensities, or their relationships with arousal. Since you cannot be sure you have drawn comprehensively from the fear and disgust categories, nor that fear and disgust are equally well conveyed through static 2-d images, any conclusions about fear and disgust per se would be a consequence of the specific images you have created. In my mind, a descriptive focus on the images does not diminish the value of the manuscript or image set at all.

Response: Thank you for the comment. We changed our focus in the revised manuscript and included only descriptions about the results we obtained. Also, we have emphasized that these results are limited to this image set.

Comment 3: The reviewer also makes a number of recommendations that should improve the impact of your work, and I encourage you to give them serious consideration. In addition, I'd like to see you add any cultural or geographic demographic information you may have on the participants who provided the ratings, given that emotion ratings are often not universal.

Response: We thank the associate editor and the reviewer for their critical comments. We have carefully considered the editor's and each reviewers' comments. We did not explicitly ask about cultural and demographic information. Thus, according to your comments, we included the following sentence stating our assumptions about the cultural and demographic backgrounds of the workers in the *Methods* (pp. 7). "*We used a crowdsourcing service in Japan. Therefore, we consider that nearly all the participants in this study were Japanese people or could understand Japanese.*"

RESPONSE TO REVIEWER 1:

Comment 1: I'm unclear why the authors chose to use a dichotomous yes this is fear vs disgust vs neither label instead of using a scale for each. In as much as disgust images can also evoke fear, it's important to know which disgust images are low threat and which are high threat. Indeed, my earlier work (March et al., 2017) found unique responses to threat vs disgust, and when I piloted images for that work, it became clear that isolating threat from disgust was important were I to make distinctions between responses to threat vs response to disgust. In as much as threat is confounded with disgust in the current work, it's hard to justify the use of the author's disgust images as purely representing disgust and not threat. Here participants are forced to choose between "disgusting" or "fearful". Had the authors used a scale for each, it would likely have been clear that some

images are disgusting AND fearful, while others are high on one but low on the other. Forcing a category onto the image may imply that one is more disgust than fearful (or vice versa), but it certainly does not imply that the image is disgust and not fearful. This is an important distinction that the authors completely ignore. If possible, I'd highly recommend they collect this data as it would make for a much more useful image set. This decision is made more confusing by the fact that the authors collected valence ratings on a continuous scale.

Comment 9: All in all, I like the paper and the image set. The authors go a bit beyond where they need to when justifying the utility of the set, and make some claims that just aren't strongly supported by their data. I'd suggest constraining the unnecessary interpretation and just present the patterns. I'd also suggest, if possible, to collect rating data of threat and disgust, separately, instead of the threat vs disgust category.

Response: We appreciate with your comment and agree with your opinion. Accordingly, we conducted an additional survey of fear and disgust ratings and included these data in the revised manuscript. We also revised the manuscript to focus on explanations about the data.

Comment 2: In their discussion of the "relationship between valence and arousal" the authors gloss over the distinction between positive and negative valence. There is a very high negative correlation between arousal and valence, and looking at Figure 2 seems to imply that arousal is highest among negative images. In fact, it looks like there is no correlation between valence and arousal for positive images.

Comment 3: The authors also state that "the relationship between valence and arousal is stronger in women than men"; I'd again like to know whether this is true for both positive and negative images. It looks like only for negative images. In the greyscale, there's an obvious group of neutrally valenced images that women but not men find highly arousing only in the grey and not in the color. This is very strange. What are these images and why would women differ? Indeed, it's unclear why any neutrally valenced image would be highly arousing.

Response: We highly appreciate this insightful comment. As you stated, we needed to carefully explain the images that were responsible for the negative correlation between valence and arousal. Accordingly, we have included the following sentences in the *Results* (pp.10), "*The OBNIS contains both emotionally negative and neutral (or relatively positive) images. Since there appears to be no correlation for the neutral valence images in the OBNIS from visual inspections, the strong negative correlation between valence and arousal ratings for images seen in this study can be attributed more to the emotionally negative images in the OBNIS.*"

We have checked the images in which women rated arousal higher in the grayscale condition than in the color condition and found that these responses occurred mainly for images of tigers and lions. We cannot be sure exactly why this difference occurred, which might be related to gender differences in the images associated with animals. Feline animals such as tigers and lions are popular as stuffed animals and animal characters among Japanese women. It is possible that when tigers' and lions' images are colored, women more than men perceive these animals as positive healing objects, leading to lower arousal ratings. However, in this paper, we decided to focus more on the description of the images than on making inferences. Therefore, we have not mentioned this possibility in the revised manuscript.

Comment 4: Figure 3 (which is really a table) is great and all, but would really benefit from being split onto multiple pages. The text is tiny. In addition, it should have the valence and arousal ratings of each image in the table.

Response: We thank the reviewer for this comment. We have included the valence and arousal ratings of each image in the Table and split the Table in to multiple pages. We also have changed the title to "Table 2" from "Figure 3."

Comment 5: On page 17, the authors mention a finding that "negative feelings experiencing when viewing disgusting images are stronger than when viewing fear images." My earlier work would disagree. In that piloting, I found that many threat images (e.g., tiger) also evoke positivity (they are cool, beautiful creatures). So it's not that people feel more negative from disgust than threat, it's just that they feel less positive. Indeed, this position is supported by the author's reporting that arousal was stronger for the fear than disgust related group.

Response: We thank the reviewer for this comment. According to your comment, we have revised the expression as follows (pp. 19), "*This result supported Soraia et al. (2018), who also reported that the disgusting images were rated more negativity than the fearful images.*"

Comment 6: I partially disagree with the statement on page 18 that characterizes fear as “high arousal” and disgust as “low arousal”. There are 25 more images in the disgust than fear set in the current work. Looking at Figure 4, at the high end of negative valence, the disgust vs fear correlation with arousal likely does not differ. It’s only when you get to the less negative or even slightly positive images that the correlation weakens. This implies that as the author’s stated, threat stimuli are high arousal, but this certainly doesn’t imply that disgust are low arousal. Looking again to Figure 4, there are plenty of high arousal disgust stimuli.

Response: We thank the reviewer for this comment. The idea that “fear” is placed at high arousal and “disgust” at low arousal is from the results of a previous study. Therefore, we think that we cannot directly come to this idea from the present data. We have revised this section to avoid misleading the reader (pp. 20) by emphasizing that arousal results of the disgust-related group are based on the images’ mean arousal ratings.

Comment 7: On page 21, the authors state that “highly arousing images tended to be consistently categorized as “fearful regardless of the valence rating. Recall my earlier point about certain threat stimuli being both positive and threat. I would suggest then that their statement is backward. It should instead read that “in this image set, high threat images tended to also be high arousal.”

P21. In their mention of the idea “that the response patterns for disgust and fear have distinct physiological, behavioral, neural, and cognitive characteristics”, I suggest they see the below referenced paper for more explanation.

Comment 8: They end the paper by focusing strongly on the idea that diffuse arousal leads to fear vs threat and base this on the previously discussed correlation. This completely missed the mark, in my mind. They really don’t need to focus on interpreting their data, but simply presenting it. They’re putting a lot of stock into people interpreting their arousal when classifying these images, and their data just don’t seem to support that position. It really just seems like a cherry-picked finding. But when you look at the figures, it doesn’t hold up.

Response: We thank the reviewer for this comment and sharing information about the paper. As the reviewer stated, we feel that our interpretations should be limited to what we could observe from the data. Therefore, we have removed this paragraph and modified the texts of the *General discussion*.